# Hierarchical amplitude modulation structures and rhythm patterns: Comparing Western musical genres, song, and nature sounds to Babytalk

Tatsuya Daikoku[1,2,3]*, Usha Goswami[1]

1 Centre for Neuroscience in Education, University of Cambridge, Cambridge, United Kingdom,
2 International Research Center for Neurointelligence, The University of Tokyo, Bunkyo City, Tokyo, Japan,
3 Center for Brain, Mind and KANSEI Sciences Research, Hiroshima University, Hiroshima, Japan

* daikoku.tatsuya@mail.u-tokyo.ac.jp

**Data Availability Statement:** All data files can be found at the following link: https://osf.io/6s8kp/. All original sound files are publicly available from the Figshare database: http://figshare.com/articles/

## Abstract

Statistical learning of physical stimulus characteristics is important for the development of cognitive systems like language and music. Rhythm patterns are a core component of both systems, and rhythm is key to language acquisition by infants. Accordingly, the physical stimulus characteristics that yield speech rhythm in "Babytalk" may also describe the hierarchical rhythmic relationships that characterize human music and song. Computational modelling of the amplitude envelope of "Babytalk" (infant-directed speech, IDS) using a demodulation approach (Spectral-Amplitude Modulation Phase Hierarchy model, S-AMPH) can describe these characteristics. S-AMPH modelling of Babytalk has shown previously that bands of amplitude modulations (AMs) at different temporal rates and their phase relations help to create its structured inherent rhythms. Additionally, S-AMPH modelling of children's nursery rhymes shows that different rhythm patterns (trochaic, iambic, dactylic) depend on the phase relations between AM bands centred on ~2 Hz and ~5 Hz. The importance of these AM phase relations was confirmed via a second demodulation approach (PAD, Probabilistic Amplitude Demodulation). Here we apply both S-AMPH and PAD to demodulate the amplitude envelopes of Western musical genres and songs. Quasi-rhythmic and non-human sounds found in nature (birdsong, rain, wind) were utilized for control analyses. We expected that the physical stimulus characteristics in human music and song from an AM perspective would match those of IDS. Given prior speech-based analyses, we also expected that AM cycles derived from the modelling may identify musical units like crotchets, quavers and demi-quavers. Both models revealed an hierarchically-nested AM modulation structure for music and song, but not nature sounds. This AM modulation structure for music and song matched IDS. Both models also generated systematic AM cycles yielding musical units like crotchets and quavers. Both music and language are created by humans and shaped by culture. Acoustic rhythm in IDS and music appears to depend on many of the same physical characteristics, facilitating learning.

SAMPH_CDS/1318572 DOI: 10.6084/m9.figshare.1318572. Please see the original article that used the speech data for more detailed information (Leong et al., 2015), for the related wiki please see https://www.cne.psychol.cam.ac.uk. All original bird song, nature sound files are available from https://www.xeno-canto.org/, https://mixkit.co/free-sound-effects/nature/, and https://www.zapsplat.com. Music and human song data has copyright, but described the detailed information in S1 Appendix.

**Funding:** This study was supported by Nakatani Foundation, JSPS KAKENHI Grant Number 21H05063 (Transformative Research Areas (B)), 22K17986 (Grant-in-Aid for Early-Career Scientists), 20K22676 (Grant-in-Aid for Research Activity Start-up), 22H05210 (Grant-in-Aid for Transformative Research Areas (A)), World Premier International Research Centre Initiative (WPI), MEXT, Japan. The sponsor played no role in the study design nor in the collection, analysis, interpretation and writing up of the data.

**Competing interests:** The authors have declared that no competing interests exist.

# 1. Introduction

The potential parallels between language and music have long fascinated researchers in cognitive science. In this paper, we examine whether a statistical learning approach previously applied to understand the development of phonology as a cognitive system in language-learning infants and children may enable theoretical advances in understanding the acoustic basis of rhythm in music. Infant language learning has been argued to begin with speech rhythm [1], and infant-directed speech (IDS), also called Babytalk or Parentese, has been described as sing-song speech. The particular prosodic or quasi-musical characteristics of IDS have been suggested to explain both natural selection for human language from an anthropological perspective [2], and to facilitate infant learning of the phonological structure of human languages [3]. Although language acquisition by human infants was once thought to require specialized neural architecture, studies of infant statistical learning have revealed that basic acoustic processing mechanisms are sufficient for infants to learn phonology (speech sound structure at different linguistic levels such as words, syllables, rhymes and phonemes; e.g. [4]). Further, the cognitive capacity of statistical learning is not restricted to verbal language, but extends to non-linguistic sounds such as tones (e.g., [5, 6]), timbres (e.g., [7, 8]) as well as rhythm and timing (e.g., [9–11]). Children who exhibit difficulties with phonological learning also exhibit rhythm processing difficulties, with both speech and musical stimuli [12]. This implies that there may be inherent common statistical properties shared by language and music, and that such statistical properties contribute to the acquisition of both language and music [13].

Modelling of the speech signal aimed at understanding the potential sensory/neural statistical properties that underpin phonological and rhythmic learning in childhood has revealed a novel set of acoustic statistics that underpin speech rhythm in infant- and child-directed speech (IDS and CDS). These novel statistics are consistent across two different modelling approaches, a spectral-amplitude modulation phase hierarchy (S-AMPH) approach based on the neural speech encoding literature [14, 15], and probabilistic amplitude demodulation (PAD, [16, 17]). The S-AMPH model was first applied to English nursery rhymes and subsequently to Babytalk [18]. The key parameter that emerged with respect to rhythm in both models and for both genres was the phase relations between a band of amplitude modulations (AMs) centred on ~2 Hz, and a band of AMs centred on ~5 Hz. When both bands peaked together, a strong syllable was heard. When a trough in the slower AM band (~2 Hz) coincided with a peak in the faster AM band (~5 Hz), a weak syllable was heard. Adult listeners' perception of vocoded English nursery rhymes could be shifted from a trochaic to iambic rhythm simply by phase-shifting the slower AM band by 180 degrees. Related experimental work using PAD showed that the phase relations between peaks and troughs in AM bands centred on ~2 Hz and ~5 Hz was critical for perceiving rhythmic metrical patterning in nursery rhymes (trochaic versus iambic, [14, 15, 18, 19]). These phase relations between peaks and troughs in AM bands centred on ~2 Hz and ~5 Hz have also been revealed by statistical modelling of other languages like Portuguese and Spanish [20, 21]. For example, Pérez-Navarro et al. [21] reported that CDS in Spanish was characterized by higher temporal regularity of the placement of stressed syllables (phase synchronization of ~2 Hz and ~5 Hz AM bands) compared to ADS in Spanish. Further, phase relations are statistical characteristics that describe music as well as language, and phase relations appear relatively uniform regarding music from different cultures [22, 23], as well as songs of different species [24]. Even prior to the acquisition of culture-specific biases of musical rhythm, infants are affected by ratio complexity [25]. Thus, phase hierarchies may be a universal aspect across music and language.

Accordingly, here we investigate the characteristics of music and child songs from the same S-AMPH modelling perspective previously applied to English, Portuguese and Spanish. In

particular, it is of interest to establish whether the phase dependency between bands of AMs centred on ~2 Hz and ~ 5 Hz will relate to musical rhythm across different genres. Theoretically, it is plausible that the physical stimulus characteristics that describe rhythm patterns in nursery rhymes and IDS may also describe the hierarchical rhythmic relationships that characterize music and child songs. According to anthropological analyses [2], it was IDS that emerged first, subsequently enabling the development of adult-directed speech (ADS, which is notably not sing-song in nature). As primitive human cultures also developed music, the same evolutionary adaptations that enabled Babytalk may underpin music as well. That is, it is possible that the AM hierarchy in music has similar structure to the AM hierarchy in IDS. The core research question addressed here is whether music will exhibit similar salient bands of AMs and similar phase dependencies between AM bands to IDS and English nursery rhymes (child-directed speech, CDS).

The theoretical framework underpinning the AM modelling approach used for CDS and Babytalk was Temporal Sampling theory (TS theory, [26]). TS theory was initially developed to explain why children with language disorders show difficulties in AM processing, with the aim of supporting musical interventions. TS theory now provides a systematic sensory/neural/cognitive framework for explaining childhood language disorders [27]. TS theory proposes that accurate sensory/neural processing of the amplitude envelope of speech is one foundation of language acquisition, and that impairments in discriminating key aspects of the envelope such as amplitude rise times at different temporal rates (which relate to speech rhythm) is one cause of developmental language disorders [28]. The amplitude envelope of any sound is the slower changes in AM (intensity or signal energy) that unfold over time. The amplitude rise time of the vowel in any syllable is a core acoustic feature related to speech rhythm [29]. Amplitude rise times are important for the perception of rhythm because they determine the acoustic experience of "P-centers." P-centers are the perceptual moment of occurrence ("perceptual center") of each musical beat or syllable for the listener [30, 31]. Amplitude rise times are typically called attack times in the musical literature [32, 33]. By TS theory, it is the rhythmic components of musical therapies for children that explain the language gains that are found, for example via the matching of the P-centers of syllable beats and musical beats [27, 34]. If musical remediation of developmental language disorders is to be optimised, then by TS theory the rhythm structures that underpin language and music should be matched at the level of physical stimulus characteristics. It is known that developmental disorders of both syntax and phonology can be helped by musical interventions [35], but related modelling of the amplitude envelope of the music used in such interventions (typically classical music) has yet to be carried out. We present such modelling of classical music and also other Western genres here.

As a control for our prediction that the AM structure of music and IDS/CDS should be highly similar, we also modelled other natural sounds that have quasi-rhythmic structure such as wind, fire, river, storms, rain, as well as non-human vocal sounds, namely birdsong. *A priori*, we expect nature sounds to have a different AM structure to IDS and CDS. Nature sounds such as rain and storms were originally used to derive PAD [16], and are characterized by AM patterns correlated over long time scales and across multiple frequency bands. However, as these sounds are not produced by humans nor shaped by human physiology and culture, there is no reason *a priori* to expect them to be similar in AM structure to IDS and CDS. Birdsong may be different, as it is more musically sophisticated and closer to human song than the other nature sounds such as wind, fire, river, storms, and rain. Indeed, a previous study revealed that the structure of nightingale rhythms, rather than other bird song rhythms such as zebra finches, are similar to the structure of human musical rhythms [24]. Therefore, we also modelled the corpus of nightingale's song studied by Roeske et al. [24]. We expected the AM patterns here to be more similar to IDS and CDS than the AM patterns for wind, rain etc. Other

approaches to modelling hierarchical temporal relations in sound signals, such as the Allan Factor approach (which detects clusters of peaks in the amplitude envelope), have suggested that thunderstorms and classical music have a similar hierarchical temporal modulation structure, we would not predict this [36]. An Allan Factor approach only reveals the overall degree of clustering found in a sound signal according to window lengths input by the modeller, which for Kello et al. [36] varied from 15 ms to 15 seconds. In contrast, the S-AMPH modelling approach utilizes the known characteristics of the human cochlea to determine its windows. Further, as noted by Kello et al. [36] themselves, an Allan Factor approach does not throw light on the relationship between individual clusters that may be identified and linguistic units. This stands in marked contrast to the S-AMPH approach to modelling infant and child language [15].

The S-AMPH model analyses the AM structure of the amplitude envelope of any sound by separating the AM characteristics from the frequency modulation (FM) characteristics. This is achieved by acoustic engineering methods for decomposing the amplitude envelope (demodulation; [17]). Demodulation approaches to characterizing the physical stimulus structure of IDS and CDS decompose the amplitude envelope into the same narrow bands imposed by the human cochlea, and then seek systematic patterns of AM [15, 18]. The AM patterns are associated with fluctuations in loudness or sound intensity, a primary acoustic correlate of perceived rhythm which is based on onset timing, beat, accent, and grouping [37]. In contrast, the FM patterns can be interpreted as fluctuations in pitch and noise [16]. Prior analyses of the average modulation spectra of Western musical genres have revealed a peak at ~2 Hz, consistent across genres like jazz, rock and classical music [38]. This is theoretically interesting, as the 2 Hz peak observed by Ding et al. [38] for music matches the modulation peak in IDS identified by S-AMPH modelling [18]. It is notable that Allan Factor modelling, which identifies nested clusters of peaks in the amplitude envelope, also finds differences between the temporal modulation structure of jazz, rock and classical music respectively. Ding et al. [38] did not find such differences in their modulation spectra approach. As the S-AMPH also takes a modulation spectra approach (but governed by knowledge about cochlear function), here we expected that the musical genres explored (which were adopted from [38]) would show a similar modulation structure to each other, as well as to IDS and CDS.

The music listener also needs to identify discrete units to gain meaning, for example musical notes and phrasing. This is analogous to infants needing to identify discrete units like syllables, words and syntactic phrases from the prosodic rhythm structure of IDS. The systematic patterns of AM nested in the amplitude envelope of both IDS and CDS, in five core spectral bands, have been demonstrated to support the identification of these discrete units. For example, application of the S-AMPH to English nursery rhyme corpora showed that the model identified 72% of stressed syllables correctly, 82% of syllables correctly, and 78% of onset-rime units correctly if a particular AM cycle was assumed to match a particular speech unit [15]. If the nursery rhymes were chanted to a regular 2 Hz beat, then the model identified over 90% of each type of linguistic unit. Accordingly, decomposition of the amplitude envelope of different musical genres may identify similar hierarchical AM structures in predictable spectral bandings that provide a perceptual basis for perceiving rhythm patterns, musical notes and musical phrasing. Whether music will exhibit similar salient bands of AMs, similar spectral banding and similar phase dependencies between AM bands to IDS and CDS is currently unknown.

It should also be noted that the modulation statistics of adult-directed speech (ADS) as revealed by the S-AMPH modelling approach are markedly different to IDS [18, 20]. ADS has significantly weaker phase synchronization between the slower bands of AMs centred on ~2 Hz and ~5 Hz compared to IDS, probably reflecting the fact that ADS is not sing-song or rhythmic. However, ADS has significantly stronger phase synchronization of bands of AMs centred on ~5 Hz and ~ 20 Hz compared to IDS. These different modulation statistics for ADS

can be interpreted as increasing the salience of acoustic information related to the phonemes in syllables [20]. The differences in statistical AM structure of the amplitude envelope of ADS vs IDS have been hypothesized to reflect the acquisition of literacy [20]. This was because the phase synchronisation between bands of AMs centred on ~2 Hz and ~5 Hz, and also ~5 Hz and ~ 20 Hz, in natural conversational adult speech increased parametrically with literacy levels (illiterate, low literate, high literate, see [20]). The acquisition of literacy remaps phonology in the human brain [39]. Music and language are ubiquitous in human societies [22], but literacy is a relatively recent cultural acquisition, so arguably the AM structure of music is more likely *a priori* to match IDS than to match ADS. Wind, rain, storms and birdsong have also all been present since early hominid times, but their statistical structure has not been constrained by the human brain. It was thus expected *a priori* that the range of nature sounds would show different statistical AM structures to music, with the possible exception of nightingale song.

Two contrasting mathematical approaches to demodulation of the amplitude envelope of music, song and nature sounds were employed, the S-AMPH [15], and PAD [16, 17]. Both models parse the amplitude envelope of the signals into an hierarchy of AM bands, but the principles underpinning their operation are different. The S-AMPH simulates the frequency decomposition known to be carried out by the cochlea [40–42], thereby aiming to decompose the amplitude envelope of music in the same way as the human ear. PAD infers the modulators and carriers in the envelope based purely on Bayesian inference, thereby carrying out amplitude demodulation on a neutral statistical basis that makes no adjustments for the human hearing system. PAD is thus a "brain-neutral" approach, but the use of Bayesian statistics means that it may reveal priors relevant to human neural learning [43]. Our expectation that the perception of musical meter may depend on the temporal alignment of AM bands centred on ~2 Hz and ~5 Hz also relates to linguistic theory [44–46]. Classically, hierarchical linguistic structures like the phonological hierarchy of prosodic, syllabic, rhyme and phoneme levels nested within speech rhythm are represented as a tree that captures the *relative prominence* of units [46, 47]. Such tree representations may also provide a good model regarding the core principles of metrical structure in music [48]. In the tree representation, a "parent" node (element) at one tier of the hierarchy encompasses one or more "daughter" nodes at a lower level of the hierarchy. The adjacent connection between the parent and daughter nodes are indicated as "branches" in the tree. To give an example from CDS, a parent node such as the trisyllabic word "pussycat" in the nursery rhyme "Pussycat pussycat where have you been," which is also the prosodic foot, would have 3 daughter nodes at the next hierarchical level, comprising the three syllables. From the prior S-AMPH modelling, the level of the prosodic foot would be derived from the cycles of AM at the ~2 Hz rate. Two AM cycles would encompass all three daughter nodes in "pussycat", while the individual syllables would be derived from the cycles of AM at the ~5 Hz rate. The phase alignment of the ~2 Hz and ~5 Hz AM cycles would then determine metrical structure. When modelled with the S-AMPH, English nursery rhymes with different metrical structures like "Jack and Jill went up the hill" (trochaic rhythm), "As I was going to St Ives" (iambic rhythm) and "Pussycat pussycat where have you been" (dactyl rhythm) all showed the same acoustic hierarchical AM structure, with three core AM bands centered on ~2 Hz, ~5 Hz, and ~20 Hz. Which metrical structure was perceived by the listener depended on the temporal alignment of AM peaks in the two slower AM bands identified by the S-AMPH, centred on ~2 Hz and ~5 Hz [14].

We note that in the previous S-AMPH research the terms "delta-rate" and "theta-rate" AM bands were adopted to describe the results of the speech demodulation analyses (see also [18]). The band of AMs centred on ~2 Hz was designated the delta-rate AM band, and the band of AMs centred on ~5 Hz was designated the theta-rate AM band. This was because TS theory was based in part on the neural oscillatory bands that track human speech in adult cortex [49–

55]. The AM bands in the speech signal revealed by the S-AMPH modelling equate temporally to electrophysiological rhythms found across the brain at the oscillatory rates of delta, theta and beta-low gamma. It is known that human speech perception relies in part on neural tracking of the temporal modulation patterns in speech at different timescales simultaneously. These temporal modulation patterns are then bound into a single speech percept, "multi-time resolution processing" [49–51, 56]. This neural tracking (also described as phase alignment, temporal alignment or entrainment) relies on acoustic components of the speech signal such as the amplitude rise times of nested AM components phase-resetting oscillatory cortical activity. In adult work, neural ("speech-brain") alignment has been shown to contribute to parsing of the speech signal into phonological units such as syllables and words [56]. For language, delta, theta, and beta/gamma oscillators in auditory cortex appear to contribute to the perception of prosodic, syllabic, and phonetic information respectively [49, 55, 57–60]. For music, oscillatory rhythms may align with rhythmic features of the acoustic input such as crotchets or musical beats [61–65]. However, possible correspondences between different oscillators and different musical units like crotchets and quavers have yet to be investigated.

Finally, there are mechanistic phase dependencies in the neural system which mirror the acoustic phase dependencies between AM bands revealed by the S-AMPH modelling of IDS, CDS and ADS. The biological evidence shows that the adjacent-band neural oscillators are not independent of, but interdependent on, each other [59, 66]. For example, the phase of delta oscillators modulates the phase of theta oscillators, and theta phase modulates beta/gamma power [59]. To date, despite a number of studies of music encompassing brain-based analyses [61, 63, 67–69], no studies have examined the temporal correlates of musical rhythm from an amplitude demodulation perspective. Our prior speech modelling suggests that it is biologically plausible to propose that rhythm perception in music and language may depend on neural entrainment to the AM hierarchies nested in the amplitude envelope of music versus IDS/CDS respectively, and that parsing of units in language and music may be an automatic consequence of neural entrainment to this hierarchy. Regarding musical signals, there are already relevant data. For example, it has been shown that neural phase locking to periodic rhythms present in musical tempi is selectively enhanced compared to frequencies unrelated to the beat and meter [65, 68]. Further, Di Liberto and colleagues revealed that musical expertise increases the accuracy of cortical tracking [62].

However, to date the amplitude envelope of different musical inputs has not been decomposed in order to discover whether beat and meter are systematically related to adjacent bands of AMs that are physically connected by mutual phase dependencies. These phase dependencies between AM bands should be consistent across different beat rates falling within each AM band, as like electrophysiological bandings the AM bands span a range of temporal rates (e.g., S-AMPH 'delta' AM band, 0.9–2.5 Hz, 'theta' AM band, 2.5–7 Hz, see Supplementary Figure, Table c in S4 Appendix). This enables the phase dependencies to be maintained across environmental variations such as speaker rate or musical tempo. Given the biological evidence that each neural oscillator modulates the adjacent-band oscillator during speech perception [59, 66], and our prior acoustic modelling data with the S-AMPH, we also hypothesized that the adjacent tiers in the temporal hierarchies of music would be highly dependent on each other compared with non-adjacent tiers, particularly for delta-theta AM coupling. By hypothesis, phase locking to different bands of AM present in the amplitude envelope of each musical genre may enable parsing of the signal to yield the perceptual experience of musical components such as minim, crotchet, and quaver (half, quarter, and eighth notes). The acoustic structure of the amplitude envelope should also contribute systematically to the perceptual experience of beat, tempo, and musical phrasing.

Note finally that our modelling approach is theoretically distinct from models that seek to identify the tactus or beat markers in singing [70], models of pulse perception based on neural resonance [71], oscillatory models of auditory attention based on dynamic attending [72], and

models of temporal hierarchical structure based on the Allan Factor approach [36, 73]. Ours is the only modelling approach to analyze the modulation structure of the amplitude envelope and further to make specific *a priori* predictions concerning expected key temporal AM rates and key hierarchical AM phase relations related to the perception of musical rhythm structure and the parsing of musical units. We predict that the phase dependency between bands of AMs centred on ~2 Hz and ~ 5 Hz will relate to musical rhythm across different genres, and that music will show similar hierarchical AM structures in predictable spectral bandings to IDS, structures that can provide a perceptual basis for perceiving musical notes and musical phrasing. The amplitude envelope is recognized as core to speech processing by speech engineers [29]. Our modelling decomposes the amplitude envelope of music instead of speech and then relates the resulting AM bands and their phase relationships to individual musical units. In principle, this approach provides a novel acoustic perspective on musical rhythm, motivated by our prior novel acoustic analyses of Babytalk.

## 2. Materials and methods

The music samples for modelling consisted of the music corpora used in the study by Ding et al. [38], with the addition of 23 children's songs in order to characterize more general properties of modulation spectra across musical genres. The final samples consisted of over 39 h of recordings (sampling rate = 44.1 kHz) of Western music (Western-classical music, Jazz, adult song, and children's songs) and musical instruments (single-voice, Violin, Viola, Cello, and Bass; multi-voice, Piano and Guitar). In addition, a range of natural sounds like birdsong (nightingale), wind and rain were extracted from sound files available on the internet (https://mixkit.co; https://www.zapsplat.com; https://www.xeno-canto.org/). The sample size and number of items in each category is provided in S1 Appendix.

The acoustic signals were normalized based on z-score (mean = 0, SD = 1). The spectro-temporal modulation of the signals was analyzed using two different algorithms for deriving the dominant AM patterns: Probability Amplitude Demodulation based on Bayesian inference (PAD; [16]) and Spectral Amplitude Modulation Phase Hierarchy (S-AMPH; [15]). The PAD model infers the modulators and a carrier based on Bayesian inference. PAD is biologically neutral and can be run recursively using different demodulation parameters each time to identify potential "priors" in the input stimulus. The S-AMPH model is a low-dimensional representation of the auditory signal, using an equivalent rectangular bandwidth ($ERB_N$) filterbank, which simulates the frequency decomposition by the cochlea [40, 42, 74]. The number and the edge of bands are determined by principal component analysis (PCA) dimensionality reduction of original high-dimensional spectral and temporal envelope representations of the input stimuli (for detail, please see Fig a in S2 Appendix). This modulation filterbank can generate a cascade of amplitude modulators at different oscillatory rates, producing the AM hierarchy. The model generates an hierarchical representation of the core spectral (acoustic frequency spanning 100–7,250 Hz) and temporal (oscillatory rate spanning 0.9–40 Hz) modulation hierarchies in the amplitude envelopes of speech and music.

### 2.1 Probability Amplitude Demodulation (PAD) model based on Bayesian inference

Amplitude demodulation is the process by which a signal ($y_t$) is decomposed into a slowly-varying modulator ($m_t$) and quickly-varying carrier ($c_t$):

$$y_t = m_t * c_t \qquad (1)$$

Probabilistic amplitude demodulation (PAD) [17] implements the amplitude demodulation as a problem of learning and inference. Learning corresponds to the estimation of the parameters that describe these distributional constraints such as the expected time-scale of variation of the modulator. Inference corresponds to the estimation of the modulator and carrier from the signals based on the learned or manually defined parametric distributional constraints. This information is encoded probabilistically in the likelihood: $P(y_{1:T}|c_{1:T}, m_{1:T}, \theta)$, prior distribution over the carrier: $p(c_{1:T}|\theta)$, and prior distribution over the modulators: $p(m_{1:T}|\theta)$. Here, the notation $x_{1:T}$ represents all the samples of the signal x, running from 1 to a maximum value T. Each of these distributions depends on a set of parameters $\theta$, which controls factors such as the typical time-scale of variation of the modulator or the frequency content of the carrier. For more detail, the parametrized joint probability of the signal, carrier and modulator is:

$$P(y_{1:T}, \ c_{1:T}, \ m_{1:T}|\theta) = P(y_{1:T}|c_{1:T}; m_{1:T}, \ \theta)*p(c_{1:T}|\theta)*p(m_{1:T}|\theta) \qquad (2)$$

Bayes' theorem is applied for inference, forming the posterior distribution over the modulators and carriers, given the signal:

$$P(c_{1:T}, \ m_{1:T}|y_{1:T}, \ \theta) = P(y_{1:T}, \ c_{1:T}, \ m_{1:T}|\theta)/P(y_{1:T}|\theta) \qquad (3)$$

The full solution to PAD is a distribution over possible pairs of modulator and carrier. The most probable pair of modulator and carrier given the signal is returned:

$$m*_{1:T}, \ c*_{1:T} = argmax \ P(c_{1:T}, \ m_{1:T}|y_{1:T}, \ \theta) \qquad (4)$$

The solution takes the form of a probability distribution which describes how probable a particular setting of the modulator and carrier is, given the observed signal. Thus, PAD summarizes the posterior distribution by returning the specific envelope and carrier that have the highest posterior probability and therefore represent the best match to the data. As noted, PAD can be run recursively using different demodulation parameters each time, thereby generating a cascade of amplitude modulators at different oscillatory rates [16]. The positive slow envelope is modelled by applying an exponential nonlinear function to a stationary Gaussian process. This produces a positive-valued envelope whose mean is constant over time. The degree of correlation between points in the envelope can be constrained by the timescale parameters of variation of the modulator (envelope), which may either be entered manually or learned from the data. In the present study, we entered the PAD parameters manually to produce modulators below 40 Hz because it is known that the core AM frequencies that contribute to speech rhythm lie below 40 Hz [15]. The carrier is interpreted as components including noise and pitches whose frequencies are much higher than the core modulation bands in phrasal, prosodic, syllabic and other phonological components. After extracting the modulators below 40 Hz, continuous Wavelet Transform (CWT) was run on each AM envelope. The procedure is depicted in Fig 1 via heat maps, which show an example of the demodulation outputs from CWT for an example of each stimulus type. Next, the demodulation outputs were normalized between 0 and 1, and averaged across all samples in each genre (instrumental music, song, and nature sounds).

## 2.2. Spectral Amplitude Modulation Phase Hierarchy (S-AMPH) model

**2.2.1. Signal processing: Spectral and temporal modulations.** This study used the same methodologies and parameters as a previous study based on CDS by Leong and Goswami [15] (for wiki, please see https://www.cne.psychol.cam.ac.uk). To establish the patterns of spectral modulation, the raw acoustic signal was passed through a 28 log-spaced $ERB_N$ filterbank spanning 100–7250 Hz, which simulates the frequency decomposition by the cochlea in a normal

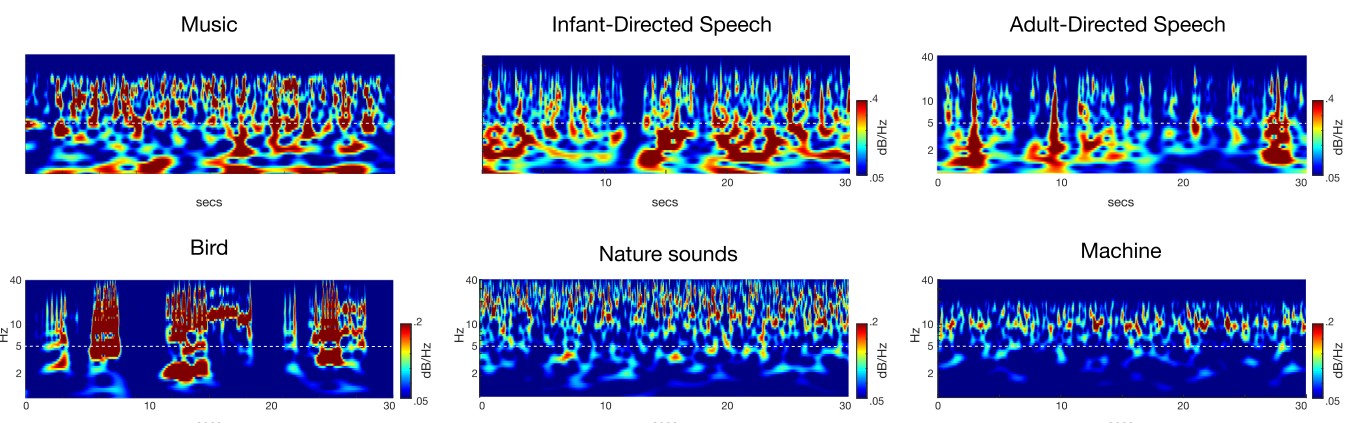

**Fig 1. Scalograms depicting the amplitude modulation (AM) envelopes derived by recursive application of PAD.** We depict music (classical), IDS (naturalistic conversation), ADS (naturalistic conversation, [75]), bird song (nightingale), nature sounds (averaged) and a man-made rhythmic sound (a machine) using Continuous Wavelet Transform (CWT), which was run on each AM envelope from randomly chosen 30-s excerpts of music, IDS, ADS, bird song, nature sounds, and machine sounds. Note that similar scalograms cannot be generated for S-AMPH because of the use of cochlear filterbanks, which means that boundary frequencies would disappear. The x-axis denotes time (30 s) and the y-axis denotes modulation rate (0.1-40Hz). The maximal amplitude is normalized to 0 dB. The demodulation outputs are shown as a heat map. It should be noted the low frequency structure (<5 Hz) visible in music and IDS is absent for the nature and machine sounds and weak for ADS and bird song. That is, systematic patches of red can be seen recurring at low frequencies for speech and music (~2 Hz and ~5 Hz), but not for nature sounds or mechanical sounds. Comparison of the temporal structures of these sounds for the low-frequency modulation rates (0–5 Hz) shows that only music and speech show strong delta- and theta-AM band patterning. The nested structure of AM patterning across the higher modulation bands (12-40Hz) is also clearly visible for the quasi-rhythmic sounds found in nature. This patterning is clearly absent for the man-made rhythmic sound of a machine.

human [40, 42]. For further technical details of the filterbank design, see Stone and Moore [76]. The parameters of the $ERB_N$ filterbanks and the frequency response characteristics are provided in S2 Appendix. Then, the Hilbert envelope was obtained for each of the 28 filtered-signals. Using the 28 Hilbert envelopes, the core spectral patterning was defined by PCA. This can identify the appropriate number and spacing of non-redundant spectral bands, by detecting co-modulation in the high-dimensional $ERB_N$ representation. To establish the patterns of temporal modulation, the raw acoustic signal was filtered into the number of spectral bands that were identified in the spectral PCA analysis. Then, the Hilbert envelope was extracted from each of the spectral bands. Further, the Hilbert envelopes of each of the spectral bands were passed through a 24 log-spaced $ERB_N$ filterbank spanning 0.9–40 Hz. Using the 24 Hilbert envelopes in each of the spectral bands, the core AM hierarchy was defined by PCA. This approach clarifies co-activation patterns across modulation rate channels.

To determine the number and the edge of the core spectral (acoustic frequency spanning 100–7,250 Hz) and temporal (oscillatory rate spanning 0.9–40 Hz) modulation bands, PCA was applied separately for spectral and temporal dimensionality reductions. PCA has previously been used for dimensionality reduction in speech studies (e.g., [77, 78]). The present study focused on the absolute value of component loadings rather than the component scores. The loadings indicate the underlying patterns of correlation between high-dimensional channels. That is, PCA loading was adopted to identify patterns of covariation between the high-dimensional channels of spectral (28 channels) and temporal (24 channels) modulations, and to determine groups (or clusters) of channels that belonged to the same core modulation bands.

**2.2.2. PCA to find the core modulation hierarchy in the high-dimensional $ERB_N$ representation.** In spectral PCA, the 28 spectral channels were taken as separate variables, yielding a total of 28 principal components. Only the top 5 principal components (PC) were considered for the further analysis, because these already cumulatively accounted for over 58% (on

average) of the total variance in the original sound signal. In temporal PCA, the 24 channels in each of the spectral bands were entered as separate variables. Only the top 3 were considered for further analysis, because these cumulatively accounted for over 55% of the total variance in the original sound signal. Each PC loading value was averaged across all samples in each genre (Western-classical music, Jazz, adult and children's song, nature sounds, birdsong) and musical instruments (single-voice: Violin, Viola, Cello, and Bass; multi-voice: Piano and Guitar). The absolute value of the PC loadings was used to avoid mutual cancellation by averaging an opposite valence across samples [14]. Then, peaks in the grand average PC loading patterns were taken to identify the core modulation hierarchy. Troughs were also identified because they reflect boundaries of edges between co-modulated clusters of channels. To ensure that there would be an adequate spacing between the resulting inferred modulation bands, a minimum peak-to-peak distance of 2 and 5 channels was set for the spectral and temporal PCAs, respectively. After detecting all the peaks and troughs, the core spectral and temporal modulation bands were determined based on the criteria that at least 2 of the 5 PCs and 1 of the 3 PCs showed a peak for spectral and temporal bands, respectively. On the other hand, the boundary edges between modulation bands were determined based on the most consistent locations of "*flanking*" troughs for each group of PC peaks that indicated the presence of a band. More detailed methodologies and examples can be found in Leong and Goswami [15] and Fig a of the S2 Appendix.

## 2.3. Mutual information between different modulation bands

We also examined whether one tier of the temporal hierarchy of music may be mutually dependent on the timing of another tier by conducting mutual information (MI) analyses. MI is a measure of the mutual dependence between the two variables. The MI can also be expressed as

$$
\begin{aligned}
I(X;Y) &= \sum_{x,y} p(x,y) \log\left(\frac{p(x,y)}{p(x)p(y)}\right) \\
&= \sum_{x,y} p(x,y) \log\left(\frac{p(x,y)}{p(x)}\right) - \sum_{x,y} p(x,y) \log p(y) \\
&= \sum_{x,y} p(x) p(y|x) \log p(y|x) - \sum_{x,y} \log p(y) p(x,y) \\
&= \sum_{x} p(x) \left(\sum_{y} p(y|x) \log p(y|x)\right) - \sum_{y} \log p(y) \left(\sum_{x} p(x,y)\right) \\
&= -\sum_{x} p(x) H(Y|X=x) - \sum_{y} p(y) \log p(y) \\
&= -H(Y|X) + H(Y) \\
&= H(Y) - H(Y|X) \ (bit)
\end{aligned}
\tag{5}
$$

where p(x,y) is the joint probability function of X and Y, p(x) and p(y) are the marginal probability distribution functions of the X and Y respectively, H(X) and H(Y) are the marginal entropies, H(X|Y) and H(Y|X) are the conditional entropies, and H(X,Y) is the joint entropy of X and Y [79].

This analysis should reveal whether a certain oscillatory rhythm X (i.e., delta, theta, alpha and beta) is dependent on another oscillatory rhythm Y. Given prior evidence regarding the interdependence of neuronal oscillatory bands [59, 66], we hypothesized that the adjacent tiers that connect via so-called "*branches*" in the AM hierarchy would be mutually dependent on each other, but non-adjacent tiers would not. If so, the results may support a hierarchical "*tree-based*" structure of musical rhythm, highlighting the applicability of an AM hierarchy to music as well as speech.

To explore this, we manually entered the PAD parameters to produce the modulators at each of five tiers of oscillatory band (i.e., delta: -4 Hz, theta: 4–8 Hz, alpha: 8–12 Hz, beta: 12–30 Hz, and gamma: 30–50 Hz) (see S3 Appendix). Note that manual entry of these parameters does not predetermine the results, rather it enables exploration of whether there is a prominent peak frequency observed in each oscillatory rate band regardless of any tempo variations (such as speeding up or slowing down) that may depend on the performer or the particular music. Accordingly, this method determines the frequencies that comprise the core temporal modulation structure of each musical genre. In each of the music samples, the modulators (envelopes) of the five oscillatory bands were converted into the frequency domain by the Fast Fourier Transform (FFT). That is, PAD is run recursively using different demodulation parameters each time, and this generates a cascade of amplitude modulators at different oscillatory rates (i.e., delta, theta, alpha, beta, and gamma), forming an AM hierarchy. We adopted the phase angle "θ" of the core temporal modulation envelopes corresponding to delta, theta, alpha and beta/gamma waves that were detected by PAD. In the S-AMPH modelling, the 5 spectral envelopes (see S2 Appendix) were passed through a second series of band-pass filters to isolate the 4 different AM bands based on the results of temporal PCA (channel edge frequencies: 0.9, 2.5, 7, 17 and 30 Hz). The phase angles were then calculated using each of the 4x5 temporal modulation envelopes. Then, using the phase angle values derived from S-AMPH and PAD respectively, the MI between different temporal modulation bands was measured.

## 2.4. Phase synchronization analyses

Based on the findings of MI, we further investigated possible multi-timescale phase synchronization between bands by computing the integer ratios between "*adjacent*" AM hierarchies (i.e., the number of parent vs. daughter elements in an AM hierarchy). This analysis addressed how many daughter elements a parent element encompasses in general in a particular musical genre. We adopted the core temporal modulation envelopes corresponding to delta, theta, alpha and beta/gamma waves detected by each of the S-AMPH and PAD modelling approaches. In the S-AMPH model, the five spectral envelopes were passed through a second series of band-pass filters to isolate the four different AM bands based on the results of the temporal PCA (channel edge frequencies: 0.9, 2.5, 7, 17 and 30 Hz). In the end, the total numbers were 4x5 temporal modulation envelopes in the S-AMPH model. In contrast, in the PAD model, we made use of the four core modulators (envelopes) corresponding to delta, theta, alpha, and beta/gamma bands, respectively.

The Phase Synchronization Index (PSI) was computed between the adjacent AM bands in the S-AMPH representation for each of the five spectral bands and in the corresponding AM bands in the PAD representation (i.e., delta vs. theta, theta vs. alpha, alpha vs. beta, beta vs. gamma phase synchronizations). The n:m PSI was originally conceptualized to quantify phase synchronization between two oscillators of different frequencies (e.g., muscle activity; Tass et al., 1998) [80], and was subsequently adapted for neural analyses of oscillatory phase-locking [81]. For example, if the integer ratio is 1:2, then the parent element encompasses 2 daughter elements for the rhythm. The PSI was computed as:

$$PSI = |e^{1(n\theta1 - m\theta2)}| \qquad (6)$$

n and m are integers describing the frequency relationship between lower and higher AM bands, respectively. An n: m ratio for each PSI was defined as n & m < 10, and 1 < n/m < 3. The values θ1 and θ2 refer to the instantaneous phase of the two AMs at each point in time. Therefore, (nθ1–mθ2) is the generalized phase difference between the two AMs, which was computed by taking the circular distance (modulus 2π) between the two instantaneous phase

angles. The angled brackets denote averaging of this phase difference over all time-points. The PSI is the absolute value of this average, and can take values between 0 and 1 (i.e., no to perfect synchronizations) [18]. A sound with a PSI of 1 is perceived as being perfectly rhythmically regular (a repeating pattern of strong and weak beats), whereas a sound with a PSI of 0 is perceived as being random in rhythm.

To investigate whether the resulting outputs truly represented systematic characteristics of natural musical rhythm, we conducted simulation analyses. We generated synthesized sounds that consisted of four temporal modulation envelopes (i.e., modulator) and one spectral frequency (carrier). That is, 2 Hz, 4 Hz, 8 Hz and 16 Hz sine waves were summarized to synthesize one compound tone waveform. The compound tone waveform was, then, multiplied by a 200 Hz sine wave. The synthesized waveform was assumed as a sound that includes temporal information of delta, theta, alpha and gamma rhythms, and spectral information of a pitch around to natural human voices. It is important to note that all of the temporal envelopes comprised simple sine waves with frequencies of a power of 2. Hence, we can hypothesize that 1:2 integer ratios should clearly and consistently appear compared with other integer ratios. If the PSIs of music show different findings from these artificial sounds, then the results may indicate that natural musical rhythm has systematic integer ratios in an AM hierarchy.

## 3. Results

### 3.1. Amplitude modulation properties of Western musical genres, song, and nature sounds from PAD

The modelling outputs from PAD are considered first, as this modelling is "brain-neutral", implementing amplitude demodulation by estimating the most appropriate modulator (envelope) and carrier based on Bayesian inference and ignoring the logarithmic frequency sensitivity of human hearing (for more detail, see Methods). Accordingly, PAD provides a good test of the hypothesis that there is a systematic hierarchy of temporal modulations underpinning both Western music and (English) IDS, but not nature sounds, with the possible exception of birdsong. Further, PAD is exempt from the possibility that the filterbank used in the S-AMPH modelling may have partially introduced artificial modulations into the stimuli through "ringing".

The PAD results are presented in Fig 2. The modelling showed that the AM bands in music matched those previously found in IDS, but the AM bands in the nature sounds did not. In particular, in panel 2d strong peaks in the delta and theta bands are clearly visible for instrumental music (red line, mean peak: delta 1.1Hz and 2.2Hz, theta 4.7Hz) and IDS (black line, mean peak: delta 1.8Hz, theta 3.3Hz), but not for nature sounds (blue line). Although the delta and theta peaks occur at slightly different temporal points, they are within close range of each other. Further, there are two matching peaks at delta and theta rates between IDS (black line in Fig 2D) and child song (light green in Fig 2D), but not in adult song, birdsong, and nature sounds. As predicted, therefore, the demodulation results for Western music match prior studies of English CDS and IDS [14, 15, 18], suggestive of shared statistical temporal characteristics of the acoustic input, to which the brain can entrain.

### 3.2. Amplitude modulation properties of Western musical genres, song, and nature sounds from S-AMPH

To investigate whether a demodulation approach based on an equivalent rectangular bandwidth ($ERB_N$) filterbank (which simulates the frequency decomposition by the cochlea) would yield similar AM bands, we applied the S-AMPH model to the same materials. We expected to

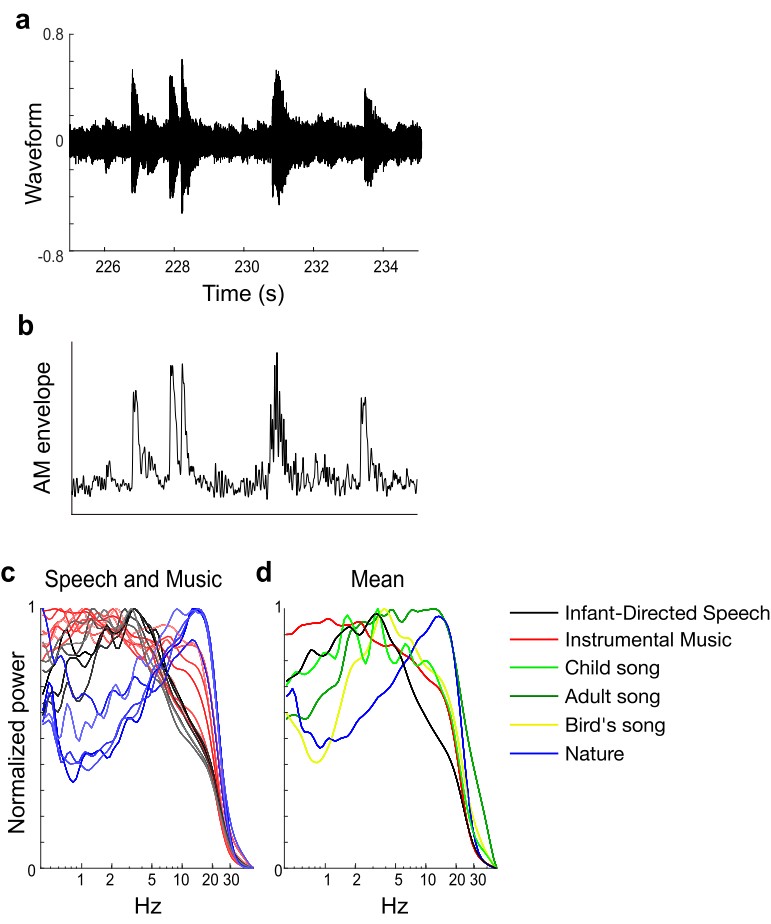

**Fig 2. Core temporal modulation rates in PAD.** The raw data (panel **a**, sound waveform of a part of the 33 Variations on a waltz by Anton Diabelli, Op. 120 by Ludwig van Beethoven) are demodulated using PAD to yield an AM envelope, shown in panel **b**. Individual lines in panels **c** represent different speakers, musical genres, and nature sounds. Panel c shows the similar acoustic statistical properties of IDS and Music: for example, hierarchical peaks in lower frequencies (~5Hz) for IDS and music, but not for nature sounds. The average of the normalized power of scalograms of Continuous Wavelet Transforms of the AM envelopes across infant-directed speech (IDS, black lines), music (red lines), nature sounds (blue lines), birdsong (yellow lines), child song (light green), and adult song (thick green) are shown in panel **d**. Panel d shows 2Hz and 5Hz peaks in both IDS and child song, but not in adult song, bird song, and nature sounds.

find a similar modulation structure to that revealed by PAD. Based on the *a priori* criteria (see Methods), the spectral PCA provided evidence for the presence of 5 core spectral bands in the spectral modulation data (300, 500, 1000, 2500 and 5500 Hz), with at least 2 out of 5 PCs showing peaks in each of these 5 spectral regions. This is shown in Fig a in S4 Appendix, which shows the grand average as well as the loading patterns and cumulative contribution ratios for each musical genre and instrument. Furthermore, we consistently observed 4 boundaries between these 5 spectral bands (350, 700, 1750 and 3900 Hz). Table a in the S4 Appendix provides a summary of these 5 spectral bands and their boundaries. It is noteworthy that these 5 spectral bands, which were consistent across musical instruments and the human voice, are proportionately-scaled with respect to the logarithmic frequency sensitivity of human hearing. As predicted, these results are similar to the spectral bands previously revealed by modelling IDS and CDS using the S-AMPH approach [14, 15, 18]. It can also be noted that the loading patterns for the 5 PCA components showed roughly similar characteristics across the genres, although there was some individual variation at each spectral modulation band (see Fig a in S4 Appendix).

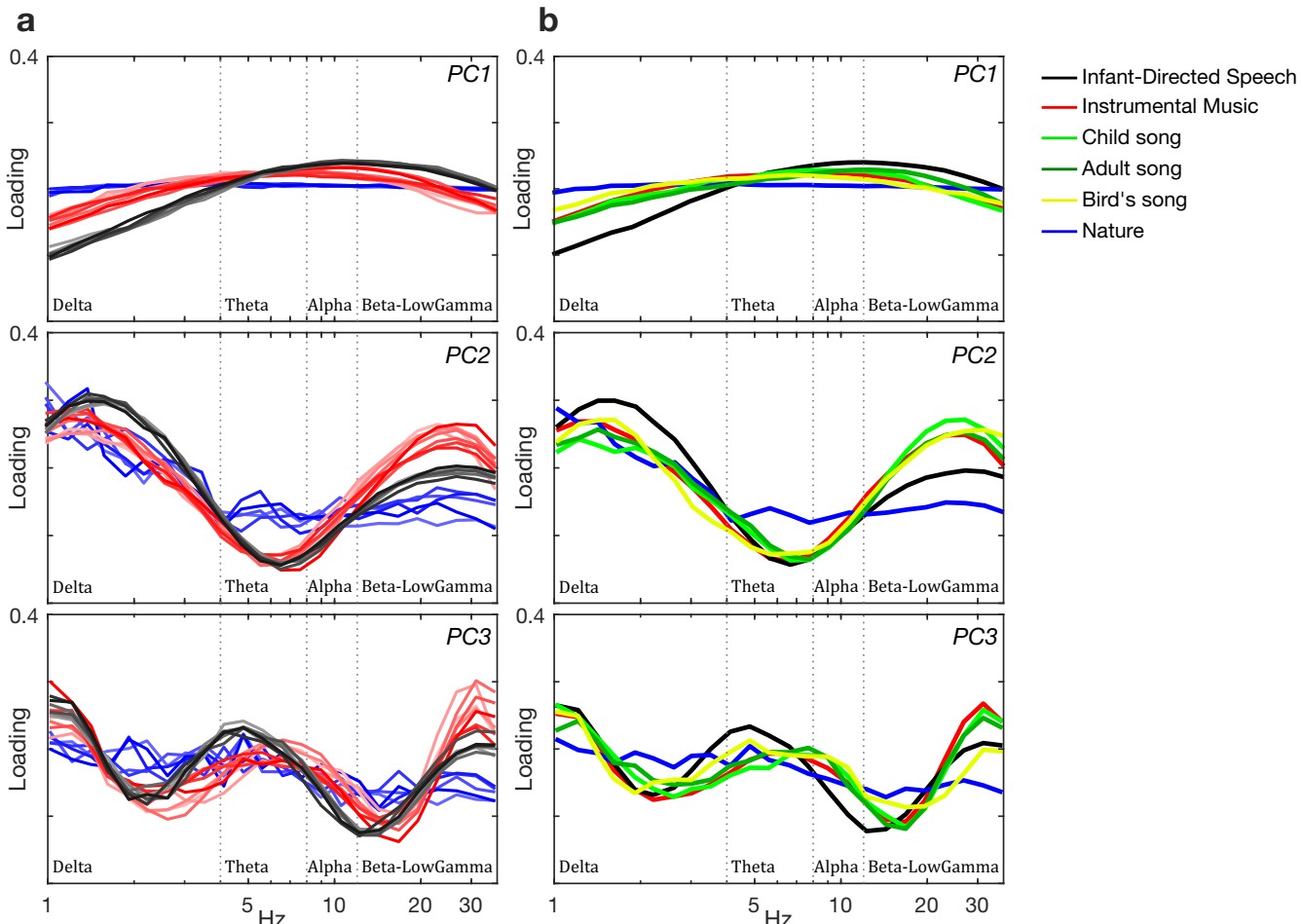

**Fig 3. Core temporal modulation rates in S-AMPH.** Grand average absolute value of the core temporal PCA component loading patterns in the S-AMPH. (**a**) Individual lines represent each of speakers (gray scale), musical genres (red scale) and nature sounds (blue scale). (**b**) Individual lines represent average of each speaker (black), music (red), child song (light green), adult song (dark green), bird song (yellow) and nature sound (blue). For detailed information about the core temporal bands and their flanking boundaries, please see Table c in S4 Appendix.

Based on the *a priori* criteria (Methods), the temporal PCA provided evidence for the presence of 4 core bands with 3 boundaries across the different musical genres and instruments. This is shown in Fig 3. These AM bands in music matched those previously found in IDS, but the AM bands in the nature sounds did not (see PC3 in Fig 3B). Further statistical detail is given in the Table b in the S4 Appendix. Fig b in S4 Appendix shows the grand average loading patterns (absolute value) for each genre for the first three principal components arising from the temporal PCA of each of the 5 spectral bands determined in the spectral PCA. Fig b in S4 Appendix also shows the temporal loading patterns and cumulative contribution ratios for each music genre and each instrument. Table c in the S4 Appendix provides a summary of these 5 spectral bands and their boundaries. Fig e in S4 Appendix shows the grand average for the modulation spectra of FFT as well as the loading patterns and cumulative contribution ratios for each music genre and instrument along with individual variation. Perceptually, cycles in these AM bands may yield the experience of crotchets, quavers, demiquavers and onsets, as shown in Table d in S4 Appendix.

In summary, the strong peaks in the delta and theta bands visible in Fig 3, along with the strong flanking trough between these bands, are clearly visible for instrumental music, human song (adult and child songs), bird song, and infant-directed speech compared to nature sounds. As predicted, therefore, the results of the temporal PCA for music essentially matches prior studies of CDS and IDS [14, 15, 18]. The one difference observed compared to the PAD modelling is for birdsong, which to the human ear (i.e., to a decomposition based on the cochlear filterbank) sounds more similar to human song, at least for the corpus of the 47 nightingale songs (see, S1 Appendix) analyzed here.

### 3.3. Mutual information in both models

To examine whether mutual dependencies between AM bands in the temporal modulation structure of different musical genres was more similar to the dependencies identified in IDS by prior S-AMPH modelling [18], a MI analysis method was employed. As noted earlier, prior modelling of IDS and CDS has revealed a significantly higher phase dependency between delta- and theta-rate AM bands compared to ADS. ADS by contrast shows a significantly higher phase dependency between theta- and beta/low gamma rate AM bands compared to IDS. Accordingly, for music we expected to find significantly higher phase dependency between delta- and theta-rate AM bands than between any other pair of AM bandings. We did not expect to find this for nature sounds. Please note that birdsong was not included in the nature sound MI analyses as the previously-presented modelling shows that the AM properties of birdsong differ with the modelling technique applied (see 3.1, 3.2 –the AM structure of bird-song is more similar to Babytalk when a model that mimics the human cochlea is utilized). For music and nature sounds (e.g., fire, river), we investigated whether higher phase dependency between delta- and theta-rate AM bands would only be detected in music, thereby matching prior studies of IDS.

When MI analyses were applied for PAD in music, four peak frequencies were detected at ~2.4 Hz, ~4.8 Hz, ~9 Hz and 16 Hz. Both delta-theta and theta-alpha mutual dependency were consistently greater than other dependencies. The MI for nature sounds looked different, with little apparent variation in MI associated with different pairings of bands. In particular, the mutual dependence between delta- and theta-rate AM bands of natural sounds was similar to non-adjacent tiers. Further detail is given in Fig d in S5 Appendix.

The MI results for the S-AMPH model showed that adjacent tiers of the AM hierarchy were mutually dependent on each other compared with nonadjacent tiers for the musical genres and for child songs. S5 Appendix shows the MI for each music genre and instrument, revealing high consistency between Western music genres, instrument and child song. Accordingly the S-AMPH model yielded similar findings to PAD, detecting peak frequencies in AM bands corresponding temporally to neural delta, theta, alpha and beta/gamma neural oscillatory bands. Further, mutual dependence between delta- and theta-rate AM bands was the strongest of all mutual dependencies detected in music, for both models. This stronger AM phase dependence matched the results of the prior speech-based modelling with IDS and CDS rather than ADS [14, 15, 18]. Accordingly, both PAD and S-AMPH MI modelling suggests that metrical structure, a feature shared by both music and speech, depends on the same core delta-theta AM phase relations in both domains.

### 3.4. Multi-timescale phase synchronization in both models

The demonstration of mutual dependency does not by itself capture metrical structure, as each AM cycle at a particular timescale may encompass one or more AM cycles at a faster timescale. To identify how many daughter elements a parent element could encompass in general, we

next investigated the integer ratios between *adjacent* AM bands. For example, if the integer ratio is 1:2, then the parent element encompasses 2 daughter elements for the rhythm. An example from speech would be a tongue twister like "Peter Piper picked a peck of pickled peppers," which follows a 1:2 ratio (two syllables in each prosodic foot). To assess the integer ratios for each pair of mutually dependent AM bands in our selected musical genres, we used PSI indices (please see S6 Appendix for the PSI for each musical genre and instrument). The PSI analyses revealed high consistency between musical genres for the phase synchronization indices generated by both S-AMPH and PAD models. Further analysis focused on the grand average (shown in Fig 4).

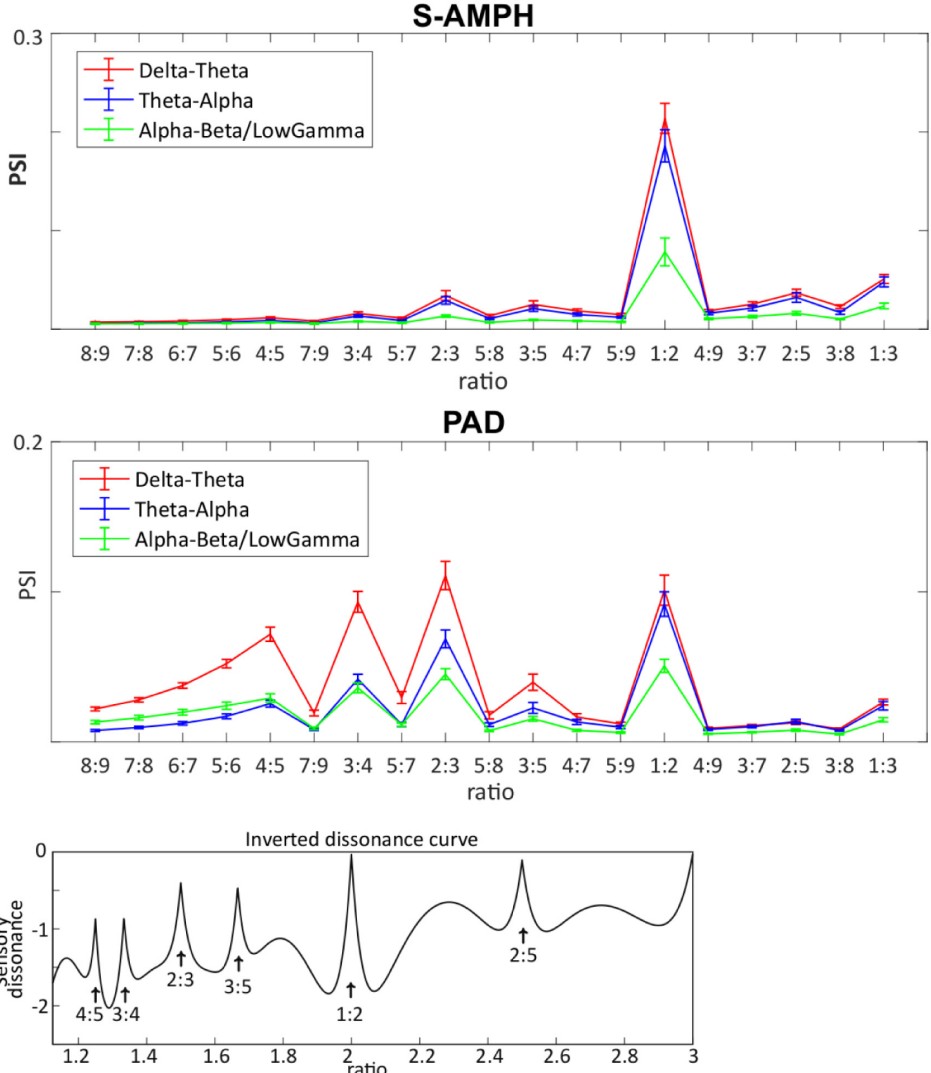

**Fig 4. Phase synchronization index between different tiers in the amplitude modulation hierarchy for music.** Both S-AMPH (a) and PAD models (b) showed that the simpler integer ratios (i.e., m/n) synchronize their phase with each other. The inverted dissonance curve **(c)** was obtained by including the first five upper partials of tones with a 440 Hz (i.e., pitch standard, A4) fundamental frequency in calculating the total dissonance of intervals [82]. It is of note that the peaks of PSI demonstrated by PAD correspond to those of the dissonance curve.

The PSI of the S-AMPH model suggested that the PSI of 1:2 integer ratios is the highest in all of the adjacent oscillatory bands. The PSIs of 1:3 and 2:3 integer ratios for the S-AMPH modelling were also higher than the other integer ratios, suggesting that the simpler integer ratios (i.e., m/n) were likely to synchronize between adjacent bands. For spoken languages, the m/n ratio between two adjacent AM bands tends to vary with linguistic factors such as how many phonemes typically comprise a syllable (e.g. 2 phonemes per syllable for a language with a consonant-vowel syllable structure like Spanish, hence a theta-beta/low gamma PSI of 1:2, but 3 phonemes per syllable for a language with largely consonant-vowel-consonant syllable structures like English, hence a theta-beta/low gamma PSI of 1:3). For music, the dominance of PSIs 1:3 and 2:3 across genres and instruments suggests more tightly controlled rhythmic dependencies than for speech.

The PSIs generated by the PAD model were similar to the S-AMPH, but PAD was more sensitive to the simple integer ratios. In PAD, the PSIs of not only the 1:2 integer ratios, but also those of the 2:3, 3:4 and 4:5 integer ratios were notably higher than the other integer ratios, particularly for the delta-theta AM band pairing (see Fig 4). The differences between models may have arisen because the filterbank used in the S-AMPH model may partially introduce some artificial modulations into the stimuli through "ringing." However, the $ERB_N$ filterbank in the S-AMPH model is the filtering process that reflects the frequency decomposition by cochlear function in the normal human ear. Hence, the different findings between S-AMPH and PAD models regarding multi-timescale phase synchronization may imply that there are differences between the physical stimulus characteristics of musical rhythm as perceived by the human brain and the purely physical and statistical structure of music.

Nevertheless, as shown in Fig 4, the PSI between delta- and theta-rate AM bands was consistently the largest PSI in both the S-AMPH and PAD models. Again, this finding is consistent with our prior findings for IDS and rhythmic CDS [15, 18]. As a further check, we also examined the PSI of sounds found in nature. The human hearing system has been receiving these quasi-rhythmic sounds at least as long as it has been receiving language and music, but unlike language and music, these sounds have not been produced by humans and shaped by human physiology and culture. Accordingly, it would not be expected that the temporal modulation structure of these natural sounds would be shared with IDS and CDS. The results showed that compared with music, the PSI between delta- and theta-rate AM bands was not consistently the largest PSI (S6 Appendix). This shows that the strong phase dependence between slower bands of AMs revealed for music and for IDS/CDS is not an artifact of the modelling approaches employed, but a core physical feature of their rhythmic structure.

Accordingly, the strong rhythmic character and acoustic temporal regularity of both infant- and child-directed speech, child song and Western music appears to be influenced by AMs in the delta band (a 2 Hz modulation peak, in music reflecting a 120 bpm rate) and by delta-theta AM phase alignment. Our modelling data for temporal frequency (i.e., "*rhythm*") also map nicely to the Plomp and Levelt [82] modeling of the dissonance curve for spectral frequency (i.e., "*pitch*") (shown in Fig 4, Bottom). This may imply that these physical properties of fast spectral frequencies are also involved in very slow temporal modulation envelopes below 40 Hz. In sum, both modelling approaches showed that the PSI of 1:2 integer ratios is the highest in all the AM band pairings, and the other simpler integer ratios (1:3, 2:3., etc) are also higher than non-integer ratios. Fig 5 provides a schematic example of the 1:2 integer ratio regarding the likely AM hierarchy in music. The figure shows in principle how musical rhythm could be hierarchically organized based on note values (i.e., crotchets, quavers, demiquavers and onsets, Fig 5, left) and the AM hierarchy (Fig 5, right).

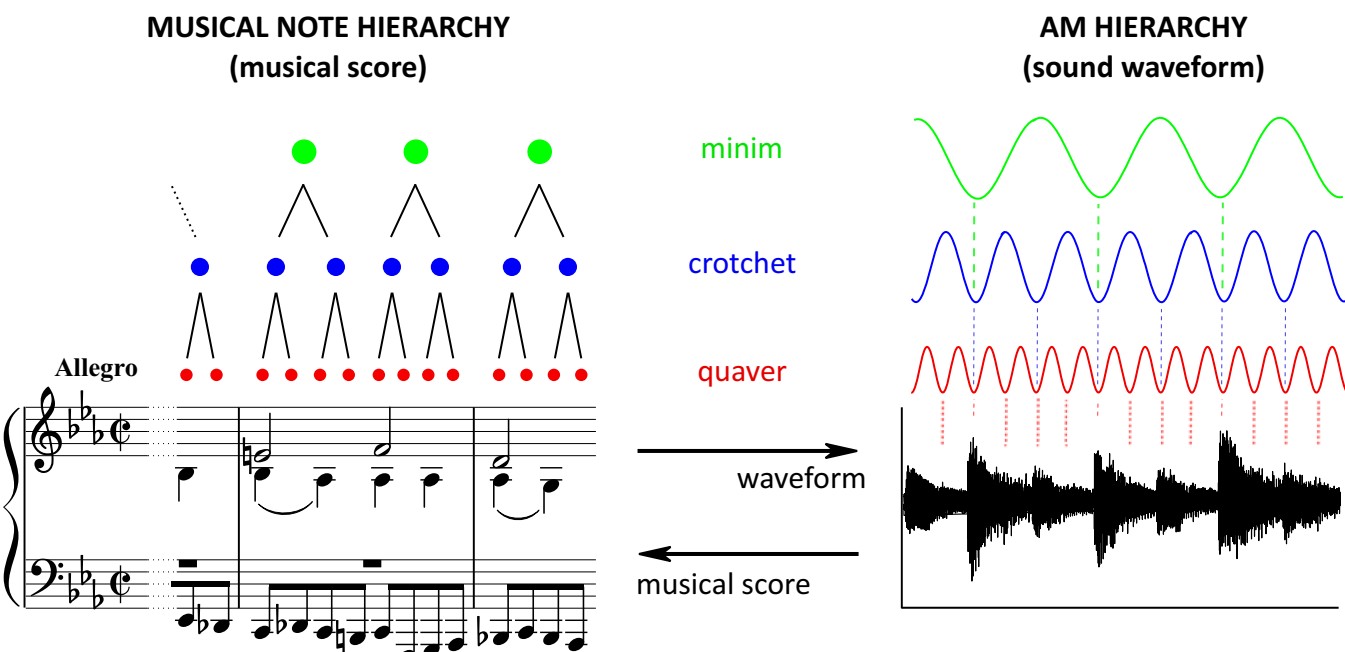

**Fig 5. Schematic depiction of the hierarchical AM structure yielding Rhythm in music.** Left and right are the representation by musical score and the corresponding sound waveform of a part of the 33 Variations on a waltz by Anton Diabelli, Op. 120 (commonly known as the Diabelli Variations) by Ludwig van Beethoven. In principle, musical rhythm could be hierarchically organized based on note values (left) matched to nested amplitude modulations (AM, right) in bandings spanning different temporal rates (for example, green ~2 Hz, blue ~4 Hz, red ~8 Hz, matching Table d in S4 Appendix). In the framework of Temporal Sampling theory, the AM bands (right) equate temporally to neural oscillatory rhythms. Auditory rhythm perception relies in part on neural tracking of the AM patterns at different timescales simultaneously (e.g., neural tracking of the green, blue, and red AMs in Fig 5 by neurophysiological delta, theta and alpha bands). This neural tracking is triggered by acoustic components of the sound signal such as the amplitude rise times (musical attack times) of the nested AM components which phase-reset oscillatory cortical activity. There are of course a large range of tempi used in music, for example slow ballads and fast dance songs. However, as shown by the black lines in the musical note hierarchy (left) and the dotted vertical lines in the AM hierarchy (right), the adjacent tiers of the hierarchy (i.e., green & blue and blue & red AM pairs) are dependent on each other compared with non-adjacent hierarchical relations (i.e., green-red AM pairing) and thus the hierarchy itself will expand or contract to fit the tempo.

### 3.5. Simulation analyses

Finally, to investigate whether the detected (dissonance curve-like) characteristics revealed by the MI and PSI analyses really represent systematic features of natural musical rhythm, we conducted simulation analyses with synthesized rhythmic but non-musical sounds. The final synthesized waveform comprised a sound that included clear rhythmic information at delta (2Hz), theta (4Hz), alpha (8Hz) and gamma (16Hz) timescales, and spectral information at a pitch around that of natural human voices (200 Hz) (for the figure, see S6 Appendix). The resulting percept was similar to a harsh rhythmic whisper. The sound is available from here https://osf.io/6s8kp/. As all of the temporal envelopes were comprised of simple sine waves with frequencies of a power of 2, PSI analyses of these artificial sounds should clearly and consistently reveal only 1:2 integer ratios compared with other integer ratios. This was the case. Thus, the simulation analyses revealed that the PSIs for natural Western musical genres were different from those for artificial rhythmic sounds. This suggests that natural musical rhythm has covert and systematic integer ratios (i.e., 2:3, 3:4 and 4:5 as well as 1:2) within the AM hierarchy, at least when considering Western musical genres.

### 4. Discussion

Here we explored the possibility that the hierarchical rhythmic (statistical AM) relationships that characterize both English Babytalk and children's nursery rhymes would also characterize

Western music [15, 18, 38]. We tested the prediction that the physical stimulus characteristics (acoustic statistics) that describe the amplitude envelope structure of IDS and CDS from a demodulation perspective would also describe Western music and child song. If child language and human music depend on the same acoustic statistics, this should facilitate initial neural learning of these culturally-determined systems. Decomposition of the amplitude envelope of IDS and CDS has previously revealed that (a) the modulation peak in IDS is ~2 Hz [18], (b) that perceived rhythmic patterning depends on three core AM bands in the amplitude envelope centred on ~2 Hz, ~5 Hz and ~20 Hz that are found systematically across the spectral range of speech [15], and (c) that varying metrical patterns such as trochaic and iambic meters can be identified by the phase relations between two of these bands of AMs (delta- and theta-rate AMs, ~2 Hz and ~5 Hz) [14]. The phase alignment (rhythmic synchronicity) of these relatively slow AM rates represents a unique statistical clue to rhythmic patterning in speech, relevant to language acquisition [83]. We predicted *a priori* that this statistical parameter (delta rate-theta rate AM phase alignment) would be present in music and human song, but not in nature sounds such as wind and rain. The physical stimulus characteristics of the amplitude envelope of different musical genres and of music produced by different instruments was expected to yield similar acoustic statistics, with classical, rock and jazz music all producing similar modulation structures. The acoustic statistics describing nature sounds were expected *a priori* to be different, as these sounds are not created by humans nor dependent on human physiology and culture. A possible exception could be the non-human-created rhythms of birdsong. Nightingale song, which has been shown to be more similar in structure to human music than other birds' song by Roeske et al. [24], was thus also modelled from a demodulation perspective.

Our demodulation analyses indeed revealed an hierarchy of temporal modulations that systematically described the acoustic properties of musical rhythm for a range of Western musical genres and instruments, as well as child song (Fig 1). Our modelling indicated highly similar acoustic statistical properties to IDS and CDS: a 2Hz modulation peak (Fig 2, panel d and Fig 3, panel b), particularly strong phase alignment between delta- and theta-rate AM bands across musical genres and human song (Mutual Information analyses), and a distinct set of preferred PSIs that indicated multi-timescale synchronization across different AM bands (Fig 4). As the brain begins learning language using IDS, and consolidates this learning via the rhythmic routines of the nursery (CDS), the present findings are consistent with the theoretical view that perceiving rhythm in both music and language may (at least early in development, prior to acquiring expertise) rely on statistical learning of the same physical stimulus characteristics. Although not tested directly here, it is likely that similar neural oscillatory entrainment mechanisms are used for encoding this hierarchical AM structure in both domains [61, 63, 67, 68]. The natural sounds analysed here have also been present since early hominid times, but their statistical structure has not been constrained by the human brain. Accordingly, learning their AM structure is less critical for human communities, and their acoustic temporal modulation structure is somewhat different to that of Babytalk and music.

Indeed, the multi-timescale synchronization found here was systematic across Western musical genres and instruments (see S6 Appendix), suggesting that this AM hierarchy contributes to building perceived rhythmic structures. The nested AM hierarchies in music may yield nested musical units (crotchets, quavers, demiquavers and onsets), just as nested AM hierarchies in CDS yield linguistic units like syllables and rhymes [15]. This possibility is depicted in Fig 5. Our modelling shows that acoustically-emergent musical units can in principle be parsed reliably from the temporal modulation spectra of the different musical genres examined, and that these units are reflected in each of delta-, theta-, alpha- and beta/gamma-rate bands of AM (Table d in S4 Appendix). To the best of our knowledge, our modelling is the first to reveal

a set of temporal statistics related to the perception of different musical units. Just as cycles of AM in IDS and CDS relate to prosodic patterns (e.g. trochaic versus iambic) and to identifying stressed syllables, syllables and rhymes, cycles of AM in music may relate to metrical structures and to units such as crotchets, quavers and demi-quavers. It is of note that these hierarchical statistical temporal dependencies should be consistent across different tempi. The dependencies refer to temporal bandings of AMs, hence the hierarchical dependencies should simply adjust to fit the tempo used in the music, for example slow ballads and fast dance songs. In similar fashion, it has been demonstrated that the hierarchical AM dependencies in speech adapt to speech rate (see [18]). Indeed, the current modelling revealed statistically strong mutual dependence (using MI estimates) between adjacent bands in the AM hierarchy across musical genres (Western classical, jazz, rock, children's songs) and musical instruments (piano, guitar, violin, viola, cello, bass, single-voice, multi-voice). This strong mutual dependence was not observed in nature sounds (shown in S5 Appendix), although birdsong was not included in these latter analyses.

In particular, regarding music the mutual dependence between delta- and theta-rate bands of AM was the strongest dependence identified by both models. Stronger mutual dependence between delta- and theta-rate AM bands could not be detected in the other nature sounds (river, fire, wind, storms, rain), even though these natural sounds are also quasi-rhythmic. The current modelling thus suggests that for Western music, delta-theta phase alignment of AM bands may underpin metrical rhythmic patterns, matching the acoustic structure of IDS and CDS. Convergent results from the phase synchronization analyses further showed that multi-timescale synchronization between delta- and theta-rate AM bands was always higher than the other PSIs regardless of the integer ratios. This was not replicated for nature sounds. The phase alignment of delta- and theta-rate bands of AM has been suggested to be a key acoustic statistic for the language-learning brain [83, 84], reflecting the placement of stressed syllables, which governs metrical patterning in speech (e.g., trochaic, iambic and dactyl meters). The present findings concerning mutual dependence and phase synchronization indicate that music may share these properties: phase alignment between delta- and theta-rate AM bands may contribute to establishing musical metrical structure as well.

Accordingly, our findings differ from a prior study using the same Western music materials, which claimed that the rhythmic properties of music and language are distinct [18]. In their speech corpora, the modulation spectrum for music peaked at 2 Hz and the modulation spectrum for speech peaked at 5 Hz. The analyses presented here suggest that the apparent dissimilarity between music and speech arises from the exclusive reliance of the speech modelling on ADS, coupled with the absence of further investigation of the AM structure of each musical genre. By contrast, our demodulation modelling approaches show better matching with temporal data from studies of IDS and CDS, where the modulation spectrum also peaks at 2 Hz (Figs 2 and 3), as well as a similar set of phase relations between AM bands (as noted, the latter were not explored by [18]). We would predict that these statistical regularities in temporal modulation may be the same for other forms of music, and for IDS and CDS in other languages, this remains to be explored. The demonstration that temporal modulation bands play a key role in rhythm hierarchies in music as well as in speech may also suggest that the same evolutionary adaptations underpin both music and language.

Another interesting result from the phase synchronization analyses regarding music was the appearance of systematic integer ratios within the AM hierarchy. These ratios were relatively uniform for nature sounds, whereas for music and child song, the 1:2 integer ratio was strongest for both models. The PSIs for 1:3 and 2:3 were also higher than the other integer ratios explored for music, for both models. For the PAD modelling approach, which does not make any adjustments for the cochlea, the 2:3, 3:4 and 4:5 integer ratios were also prominent.

This statistical patterning appears to reflect the tightly-controlled rhythmic dependencies in music, and may offer an acoustic model for capturing the different metrical structures and integer ratios that characterize music from different cultures [22, 23], as well as the songs of different species [24]. For example, even prior to the acquisition of culture-specific biases of musical rhythm, young infants (5-month-olds) are influenced by ratio complexity [25]. Our modelling further suggests that the AM bands in music are related by integer ratios in a similar way to the integer ratios relating notes of different fundamental frequencies that create harmonicity (see the similarity between the PSIs for the two models shown in Fig 4 and the dissonance curve measured by Plomp & Levelt, [82]). Converging prior modelling of speech has shown that the probability distribution of amplitude–frequency combinations in human speech sounds relates statistically to the harmonicity patterns that comprise musical universals [85]. Our modelling appears to suggest that the simple integer ratios (i.e., 1:2, 1:3, and 2:3) in the AM hierarchy comprise a fundamental set of statistics for musical rhythm perception. This fits well with prior data from Jacoby and McDermott [86], who demonstrated that certain integer ratios are prominent across music from both Western and non-Western cultures. Our acoustic modelling suggests that AM phase hierarchies may play as strong a role as harmonicity regarding universal aspects of human hearing that are important for both music and language.

The modelling presented here also converges conceptually with past studies designed to detect pulse based on neural resonance theory [71]. Pulse is the perceptual phenomenon in which an individual perceives a steady beat. Large et al. [71] suggested that the perception of pulse emerges through nonlinear coupling between two oscillatory networks, one representing the physical properties of the stimulus and a second network that integrates inputs from the sensory system. The nonlinear interactions between the two give rise to oscillatory activity not only at the frequencies present in the physical stimulus, but also at more complex combinations, including the pulse frequency. Consistent with this view, Tal et al. [87] reported phase locking for the adult brain at the times of a missing pulse, even though the pulse was absent from the physical stimulus. This suggests that neural activity at the pulse frequency is (for adults) internally generated rather than being purely stimulus-driven. From this perspective, our modelling (i.e., S-AMPH and PAD) is capturing the physical stimulus characteristics (the modulation structure of the amplitude envelope and its internal phase relations) rather than capturing internally-generated oscillatory activity. To our knowledge, missing pulse phenomena have not yet been studied in infants. It may be that early learning of hierarchical phase relations from the amplitude envelopes of musical inputs may be required for the internal generation of missing pulse phenomena. On the other hand, ERP studies show that even newborns can detect beat violations in oddball paradigms, where occasionally a deviant rhythm with a missing downbeat is heard in place of a standard metrical rhythm [88]. Further studies with infants may also be able to investigate the phase relationships between missing pulses or beats and higher hierarchical units such as musical phrasing or prosody.

The modelling presented here is also relevant to the remediation of childhood language disorders. The possible utility of musical interventions for children with disorders of language learning such as developmental language disorder (DLD) and developmental dyslexia has long been recognized [35, 89–91]. Such interventions are likely to be most beneficial when the temporal hierarchy of the music corresponds to the temporal hierarchy underpinning speech rhythm [27, 83]. Careful consideration of the statistical rhythm structures characterizing speech in different languages may thus lead to better remedial outcomes. For example, our findings suggest that for children with disorders of English language learning, interventions using Western music should be beneficial via the shared temporal hierarchy with English IDS and CDS. Further, it is possible that such interventions could be beneficial for second language

learners. A caveat is that here we modelled musical genres that could be designated WEIRD corpora (originating from Westernized, educated, industrialized, rich and democratic societies). Accordingly, further studies are necessary to understand how music interventions can contribute to improving speech processing in other languages.

In conclusion, the present study revealed that the acoustic statistics that describe rhythm in music from an amplitude envelope decomposition perspective match those that describe IDS and CDS. The physical stimulus characteristics that describe nature sounds are different. The modelling demonstrates a core acoustic hierarchy of AMs that yield musical rhythm across the amplitude envelopes of different Western musical genres and instruments, with mutual dependencies between AM bands playing a key role in organizing rhythmic units in the musical hierarchy for each genre. Accordingly, biological mechanisms that exploit AM hierarchies may underpin the perception and development of both language and music. In terms of evolution, the novel acoustic statistics revealed here could also explain cross-cultural regularities in musical systems [23]; this remains to be tested.

## Supporting information

**S1 Appendix. Corpora of music, speech, and nature sounds.**
(DOCX)

**S2 Appendix. Signal processing steps in S-AMPH model.**
(DOCX)

**S3 Appendix. Signal processing steps in PAD model.**
(DOCX)

**S4 Appendix. Individual variation of PCA loadings in S-AMPH model and those of FFT in the PAD model.**
(DOCX)

**S5 Appendix. Individual variation of mutual information.**
(DOCX)

**S6 Appendix. Individual variation of PSI in each integer ratio.**
(DOCX)

## Author Contributions

**Conceptualization:** Usha Goswami.

**Formal analysis:** Tatsuya Daikoku.

**Funding acquisition:** Tatsuya Daikoku.

**Investigation:** Tatsuya Daikoku.

**Methodology:** Tatsuya Daikoku, Usha Goswami.

**Supervision:** Usha Goswami.

**Visualization:** Tatsuya Daikoku.

**Writing – original draft:** Tatsuya Daikoku.

**Writing – review & editing:** Tatsuya Daikoku, Usha Goswami.

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
