## [Decision Letter · Decision Letter 0]

2 Mar 2022

PONE-D-21-38111Hierarchical Amplitude Modulation Structures and Rhythm Patterns: Comparing Western Musical Genres, Song, and Nature Sounds to BabytalkPLOS ONE

Dear Dr. Daikoku,

Thank you for submitting your manuscript to PLOS ONE. After careful consideration, we feel that it has merit but does not fully meet PLOS ONE’s publication criteria as it currently stands. Therefore, we invite you to submit a revised version of the manuscript that addresses the points raised during the review process.

Two expert reviewers have reviewed your submission. While both reviewers acknowledged that this manuscript addresses a timely and interesting question, they raised a bunch of issues related to the hypothesis, methodology and selection of testing materials. They also asked about the language specificity issue, and questioned to what extent the findings can generalize to languages other than English. I'd encourage you to incorporate the comments from the two reviewers into your revision as much as possible.

We look forward to receiving your revised manuscript.

Kind regards,

Caicai Zhang

Academic Editor

PLOS ONE

Journal Requirements:

3. PLOS requires an ORCID iD for the corresponding author in Editorial Manager on papers submitted after December 6th, 2016. Please ensure that you have an ORCID iD and that it is validated in Editorial Manager. To do this, go to ‘Update my Information’ (in the upper left-hand corner of the main menu), and click on the Fetch/Validate link next to the ORCID field. This will take you to the ORCID site and allow you to create a new iD or authenticate a pre-existing iD in Editorial Manager. Please see the following video for instructions on linking an ORCID iD to your Editorial Manager account: https://www.youtube.com/watch?v=_xcclfuvtxQ.

“This study was supported by Nakatani Foundation, JSPS KAKENHI Grant Numbers 20K22676 (Research Activity Start-up), 21B101(Transformative Research Areas), and World Premier International Research Centre Initiative (WPI), MEXT, Japan. The sponsor played no role in the study design nor in the collection, analysis, interpretation and writing up of the data.”

“This study was supported by Nakatani Foundation, JSPS KAKENHI Grant Numbers 20K22676 (Research Activity Start-up), 21B101(Transformative Research Areas), and World Premier International Research Centre Initiative (WPI), MEXT, Japan. The sponsor played no role in the study design nor in the collection, analysis, interpretation and writing up of the data.”

“NO authors have competing interests.”

Reviewers' comments:

Reviewer's Responses to Questions

**Comments to the Author**

1. Is the manuscript technically sound, and do the data support the conclusions?

Reviewer #1: Partly

Reviewer #2: Yes

2. Has the statistical analysis been performed appropriately and rigorously? 

Reviewer #1: I Don't Know

Reviewer #2: Yes

3. Have the authors made all data underlying the findings in their manuscript fully available?

Reviewer #1: Yes

Reviewer #2: No

4. Is the manuscript presented in an intelligible fashion and written in standard English?

Reviewer #1: Yes

Reviewer #2: Yes

5. Review Comments to the Author

Reviewer #1: Although the basic question raised by the paper is valid and interesting, it can be formulated in a more logically sound manner, and it seems that the research question has been based on more presumptions than it should. Here are some questions for the authors to address further: (1) although music and language share many overlapping features, but one essential difference is the pitch variation. In non-tonal language such as English, there is no pitch variation that can match that in the music. Therefore, to set out by directly "transplanting" the statistical learning approach to understand the rhythm in music is questionable. A potentially more objective way to start might be to examine some simple rhythmic patterns (without pitch, e.g., monotone duple/triple metre) first. (2) why the statistical learning approaches that work in explaining linguistic rhythm would lead to the authors expecting a comparable results in explaining the musical rhythm in the first place? It seems not that convincing and more literature and logical flow should be added there. (3) using the many quasi-rhythmic sounds as the control was a nice try, however, birdsong can be more musically sophisticated than other sounds such as the rain or wind, why it should be counted as quasi-rhythmic? why choosing nightingale birdsong as the representative for the birdsong? More explanations are needed.

Reviewer #2: Daikoku and Goswami present two computational approaches to analyze rhythmic patterns and hierarchical amplitude modulation structures in different acoustic materials. They built on previous work and extend here the analyses approach to more divers acoustic materials (music, song, nature sounds, babytalk) and two analyses approaches (S-AMPH, PAD).

This manuscript contributes to a timely and rapidly growing research domain and will stimulate new experimentations and perspectives for the investigation of typical and atypical functioning as well as the rehabilitation thereof. The potential impact of this contribution could be enhanced by considering the following extensions and revisions, as well as by making available their two analyzing approaches and programs via Open Science (or on reasonable request), which would allow interested research groups to further explore and test their behavioral and neural relevance. The discussion section could also further gain in impact by proposing more concrete testing hypothesis, e.g., what kind of specific music/language material matching could be used for training and/or cueing, for example.

- The authors propose a matching between AM cycles and musical units like crotchets or quavers (e.g., figure 5). Considering the large range of tempo used in the musical repertoire, such as slow ballads and fast dance songs, which contrast from the smaller tempo range for nursery rhymes, the proposed matching needs some further explanation and/or restricted to an illustrative example and/or be removed. Indeed, it does not seem straightforward how the quarter level will be matched between a musical piece at 60 bpm versus 130 bpm.

- The figures and descriptions of the findings (also in comparison to previous work) reveals that it would be interesting to extend the present set of analyses also to adult-directed speech in addition. This would allow for directly integrating the present work in previous approaches and allow for additional perspectives for future research.

- Relatedly, the authors discuss potential differences between different types of music (e.g., from different cultures with different underlying metric structures) as well as different languages (e.g., page 36 and elsewhere). It would be great to see how the models react to musical pieces in even versus uneven meters as well as to speech excerpts of different languages.

- Figure 2 suggests two slightly shifted peaks for music vs speech (e.g., with maxima of ˜1.5 vs. ˜2Hz?). Please add the exact maxima in the text.

The introduction includes a presentation of the Temporal Sampling theory. This section could be extended and clarified, notably how it links to other research domains focusing on the potential role of oscillations in this processing (whether typical or atypical). The explanation should be extended from the amplitude rise time of the vowel to that of a consonant, which contains the onset part and should be particularly relevant for the extraction of timing information. Regarding potential musical interventions and the discussion of amplitude envelopes of music, the authors should clarify the potential mechanisms that are boosted by the interventions.

Regarding the window lengths selected by the modeler for the Allan Factor approach, it would be interesting to specify the time windows used (time scales) and whether this approach would work also for longer utterances? (page 7)

The authors did a great job in explaining TS, AM and brain oscillations (Page 12), but the explanations could be clarified by adding a figure illustrating the different elements (e.g., page 11).

Tal et al. (2017) addressing the ‘missing pulse’ phenomenon might be an interesting reference in the present context too. How do the present modeling approaches handle situations of the missing beat? Or more generally of syncopation and groove? The work by E. W. Large (University of Connecticut) proposes some interesting modeling perspectives for rhythm, meter and temporal information (as well as tonality), including the implication of neural oscillations as well as entrainment. It would be relevant for the present approach and for the integration in the research domain to also address these research approaches.

Methods

Page 15:

- Which are the “a priori assumptions” here?

- Did the S-AMPH model use the same parameters as in Leong et al?

Page 19:

- Does Figure 1 present just one example for music, IDS and machine? Could the authors propose a summary across items and show frequency ranges (in an additional figure?), aiming to provide information about the generalization of the observed pattern.

- Figure 1 proposes a comparative presentation of the four categories only for PAD. It would be informative for the reader to see the same type of presentation for S-AMPH.

Page 27: “suggestive of shared physical stimulus characteristics to which the brain can entrain” – So this would concern only the evoked responses (entrainment based on the input), can the authors also address entrainment based on cognitive construct (such as metrical hierarchies) that are not necessarily implemented in the acoustic signal?

Figure 3: How many items were used for each category here?

Discussion

Page 46: Please clarify the extension to music intervention. How could the findings be linked? Which type of music or characteristics should be used to train speech (across development as well as across different languages)?

Additional comments:

Abstract:

Please clarify the wording, notably to which part of the sentence “which matched IDS” is referring to.

Introduction:

Page 4: this cognitive capacity has been shown to go beyond verbal material, that is, it is not restricted to language, but extends to non-linguistic materials, such as tones (e.g., Saffran et al., 1999; Tillman & Poulin-Charronnat, 2010), timbres (e.g., Loui et al., 2022; Tillman & McAdams, 2003; Tillman & Hoch, 2010) as well as rhythm and timing (e.g., Prince et al., 2018; Brandon et al., 2012).

Page 5: Does this depend on the language? Here the authors refer to their own work in English (Leong et al). It would be interesting to comment on extensions to other languages (e.g., German, French, Spanish, Italian) and whether it would be affected by differences between American English, British English, Australian ..?

Page 13. A cortical tracking approach has been applied to music (even though not specifically to rhythm) by Pelofi, Shamma, and collaborators (see https://clame.nyu.edu/scientists/claire-pelofi), and which might be of interest for the authors here too.

Page 30 “the nightingale song” – this suggests as if only 1 item was used here. Please clarify (and if yes, justify) or, preferably, extend to more items per category.

Page 32: “When MI analyses were applied …..” Does this refer to the music material? Please clarify.

Page 40 “similar to a harsh rhythmic whisper” For the interested reader, it would be great to add a sound file to the supplementary material section.

Page 42: The text refers here to “across Western musical genres and instruments” while most figures do not separate across genres. It would be helpful to see how findings potentially change as well as regarding whether the musical excerpts include voice or not.

Appendix 1 lists the used materials - would the sound files be available upon (reasonable) request?

6. PLOS authors have the option to publish the peer review history of their article (what does this mean?). If published, this will include your full peer review and any attached files.

Reviewer #1: No

Reviewer #2: No

---

## [Author Response · Author response to Decision Letter 0]

11 May 2022

RESPONSE TO REVIEWERS 1: 

I am extremely grateful for the Reviewers’ insightful comments on our manuscript. We have studied the Reviewers’ comments carefully and have made the necessary corrections in our paper. We believe that the comments have helped us significantly improve and refine the manuscript. Our responses to the Reviewers’ comments are as follows.

Comment 1:

Although music and language share many overlapping features, but one essential difference is the pitch variation. In non-tonal language such as English, there is no pitch variation that can match that in the music. Therefore, to set out by directly "transplanting" the statistical learning approach to understand the rhythm in music is questionable. A potentially more objective way to start might be to examine some simple rhythmic patterns (without pitch, e.g., monotone duple/triple metre) first. 

Response: I thank the reviewers for the pertinent comments. However, pitch variation is independent of rhythm variation, as indeed demonstrated by our modelling. The important research question of the current study is to understand “temporal” (but not spectral, like pitch) structure. Infant language learning has been argued to begin with speech rhythm (Mehler et al., 1988), and infant-directed speech (IDS), also called Babytalk or Parentese, has been described as sing-song speech. Spectral features such as pitch variation have been widely examined in IDS, as well as music (Schwarz, Howe & Purves, 2003). Our relatively new topic (i.e., temporal feature of sound waveform) is becoming more and more important in the fields of both speech processing and music. As noted, we did examine the spectral features that carry temporal rhythm within our framework of amplitude modulation (see Fig b in S2 Appendix). The results were shown in fig b of S4 Appendix. The modelling showed that the spectral component does not affect temporal rhythm.

Comment 2:

Why the statistical learning approaches that work in explaining linguistic rhythm would lead to the authors expecting a comparable results in explaining the musical rhythm in the first place? It seems not that convincing and more literature and logical flow should be added there. 

Response: Thank you for the comment. To address this, we have restructured parts of the Introduction accordingly. Although language acquisition by human infants was once thought to require specialized neural architecture, studies of infant statistical learning have revealed that basic acoustic processing mechanisms are sufficient for infants to learn phonology. Further, the cognitive capacity of statistical learning is not restricted to verbal language, but extends to non-linguistic sounds, such as tones (e.g., Saffran et al., 1999; Francois et al., 2011), timbres (e.g., Loui et al., 2022; Tsogli et al., 2019) as well as rhythm and timing (e.g., Prince et al., 2018; Brandon et al., 2012; Daikoku et al., 2020). Children who exhibit difficulties with phonological learning also exhibit rhythm processing difficulties, with both speech and musical stimuli (Goswami, 2015, for review). This implies the inherent common statistical properties between language and music. We explained these points in the Introduction section as follows. 

“Although language acquisition by human infants was once thought to require specialized neural architecture, studies of infant statistical learning have revealed that basic acoustic processing mechanisms are sufficient for infants to learn phonology (speech sound structure at different linguistic levels such as words, syllables, rhymes and phonemes; e.g. Saffran, 2001). Further, the cognitive capacity of statistical learning is not restricted to verbal language, but extends to non-linguistic sounds such as tones (e.g., Saffran et al., 1999; Francois et al., 2011), timbres (e.g., Loui et al., 2022; Tsogli et al., 2019) as well as rhythm and timing (e.g., Prince et al., 2018; Brandon et al., 2012; Daikoku et al., 2020). Children who exhibit difficulties with phonological learning also exhibit rhythm processing difficulties, with both speech and musical stimuli (Goswami, 2015, for review). This implies that there may be inherent common statistical properties shared by language and music, and that such statistical properties contribute to the acquisition of both language and music (Politimou et al., 2019).”

“Theoretically, it is plausible that the physical stimulus characteristics that describe rhythm patterns in nursery rhymes and IDS may also describe the hierarchical rhythmic relationships that characterize music and child songs. According to anthropological analyses (Falk, 2004), it was IDS that emerged first, subsequently enabling the development of adult-directed speech (ADS, which is notably not sing-song in nature). As primitive human cultures also developed music, the same evolutionary adaptations that enabled Babytalk may underpin music as well. That is, it is possible that the AM hierarchy in music has similar structure to the AM hierarchy in IDS. The core research question addressed here is whether music will exhibit similar salient bands of AMs and similar phase dependencies between AM bands to IDS and English nursery rhymes (child-directed speech, CDS).”

Comment 3:

Using the many quasi-rhythmic sounds as the control was a nice try, however, birdsong can be more musically sophisticated than other sounds such as the rain or wind, why it should be counted as quasi-rhythmic? why choosing nightingale birdsong as the representative for the birdsong? More explanations are needed.

Response: I thank the reviewers for the pertinent comments. To address this, we have explained in more detail why we chose nightingale songs. A priori it could either be argued that birdsong will differ from human music (our original argument), or that it will be more similar to human music than other nature sounds for the reasons pointed out by the reviewer. We now acknowledge both possibilities. In some of the analyses, we indeed detected similarity of AM structure between human voice and birdsong, as shown in Fig 3. In addition, we choose nightingales as the representative case for birdsong because a previous study revealed that nightingale rhythms, rather than another bird song rhythm such as zebra finches are most similar to human musical rhythms (Roeske et al., 2020). We revised this part in whole manuscript as follows.

“Abstract: Quasi-rhythmic and non-human sounds found in nature (birdsong, rain, wind) were utilized for control analyses.”

“Introduction: As a control for our prediction that the AM structure of music and IDS/CDS should be highly similar, we also modelled other natural sounds that have quasi-rhythmic structure such as wind, fire, river, storms, rain, as well as non-human vocal sounds, namely birdsong. A priori, we expect nature sounds to have a different AM structure to IDS and CDS. Nature sounds such as rain and storms were originally used to derive PAD (Turner, 2010), and are characterized by AM patterns correlated over long time scales and across multiple frequency bands. However, as these sounds are not produced by humans nor shaped by human physiology and culture, there is no reason a priori to expect them to be similar in AM structure to IDS and CDS. Birdsong may be different, as it is more musically sophisticated and closer to human song than the other nature sounds such as wind, fire, river, storms, and rain. Indeed, a previous study revealed that the structure of nightingale rhythms, rather than other bird song rhythms such as zebra finches, are similar to the structure of human musical rhythms (Roeske et al., 2020). Therefore, we also modelled the corpus of nightingale’s song studied by Roeske et al. (2020). We expected the AM patterns here to be more similar to IDS and CDS than the AM patterns for wind, rain etc.” 

RESPONSE TO REVIEWERS 2: 

I wish to express our strong appreciation to the Reviewer for the insightful comments on our manuscript. We have studied the Reviewer’s comments very carefully and have made necessary corrections. We feel the comments have helped us significantly improve the manuscript.

Comment 1:

Daikoku and Goswami present two computational approaches to analyze rhythmic patterns and hierarchical amplitude modulation structures in different acoustic materials. They built on previous work and extend here the analyses approach to more divers acoustic materials (music, song, nature sounds, babytalk) and two analyses approaches (S-AMPH, PAD). This manuscript contributes to a timely and rapidly growing research domain and will stimulate new experimentations and perspectives for the investigation of typical and atypical functioning as well as the rehabilitation thereof. The potential impact of this contribution could be enhanced by considering the following extensions and revisions, as well as by making available their two analyzing approaches and programs via Open Science (or on reasonable request), which would allow interested research groups to further explore and test their behavioral and neural relevance. The discussion section could also further gain in impact by proposing more concrete testing hypothesis, e.g., what kind of specific music/language material matching could be used for training and/or cueing, for example.

Response: I thank the reviewers for all of the pertinent comments. Based on all of the comments, we thoroughly revised the manuscript. We also provided more concrete testing hypothesis based on the comment. We are sure that the manuscript improved very much. Further, as indicated, we provided all of the data analyzed via Open Science at the following link (https://osf.io/6s8kp/). In addition, all original sound files are publicly available from the Figshare database: http://figshare.com/articles/SAMPH_CDS/1318572 DOI: 10.6084/m9.figshare.1318572. Please see the original article that used the speech data for more detailed information (Leong et al., 2015). All original bird song, nature sound files are available from https://www.xeno-canto.org/, https://mixkit.co/free-sound-effects/nature/, and https://www.zapsplat.com. Music and human song data has copyright, but described the detailed information in S1 Appendix. We described them in the Data Availability section.

Comment 2:

The authors propose a matching between AM cycles and musical units like crotchets or quavers (e.g., figure 5). Considering the large range of tempo used in the musical repertoire, such as slow ballads and fast dance songs, which contrast from the smaller tempo range for nursery rhymes, the proposed matching needs some further explanation and/or restricted to an illustrative example and/or be removed. Indeed, it does not seem straightforward how the quarter level will be matched between a musical piece at 60 bpm versus 130 bpm.

Response: I thank the reviewers for the pertinent comments. Our intention was not to claim that the quarter notes were always at 4 Hz, rather that the phase relations in the hierarchy could be matched to notes at different temporal levels in a piece of music within the AM bands described by our modelling. We have revised fig 5 accordingly and altered the corresponding part in the manuscript as follows.

“Discussion section: It is of note that these hierarchical statistical temporal dependencies should be consistent across different tempi. The dependencies refer to temporal bandings of AMs, hence the hierarchical dependencies should simply adjust to fit the tempo used in the music, for example slow ballads and fast dance songs. In similar fashion, it has been demonstrated that the hierarchical AM dependencies in speech adapt to speech rate (see Leong et al., 2017). Indeed, the current modelling revealed statistically strong mutual dependence (using MI estimates) between adjacent bands in the AM hierarchy across musical genres (Western classical, jazz, rock, children’s songs) and musical instruments (piano, guitar, violin, viola, cello, bass, single-voice, multi-voice).”

“Result section (3.4. Multi-Timescale Phase Synchronization in Both Models): Fig 5 provides a schematic example of the 1:2 integer ratio regarding the likely AM hierarchy in music. The figure shows in principle how musical rhythm could be hierarchically organized based on note values (i.e., crotchets, quavers, demiquavers and onsets, Fig 5, left) and the AM hierarchy (Fig 5, right).”

“Fig 5. Schematic Depiction of the Hierarchical AM Structure yielding Rhythm in Music. Left and right are the representation by musical score and the corresponding sound waveform of a part of the 33 Variations on a waltz by Anton Diabelli, Op. 120 (commonly known as the Diabelli Variations) by Ludwig van Beethoven. In principle, musical rhythm could be hierarchically organized based on note values (left) matched to nested amplitude modulations (AM, right) in bandings spanning different temporal rates (for example, green ~2 Hz, blue ~4 Hz, red ~8 Hz, matching S4 Appendix Table d). In the framework of Temporal Sampling theory, the AM bands (right) equate temporally to neural oscillatory rhythms. Auditory rhythm perception relies in part on neural tracking of the AM patterns at different timescales simultaneously (e.g., neural tracking of the green, blue, and red AMs in Fig 5 by neurophysiological delta, theta and alpha bands). This neural tracking is triggered by acoustic components of the sound signal such as the amplitude rise times (musical attack times) of the nested AM components which phase-reset oscillatory cortical activity. There are of course a large range of tempi used in music, for example slow ballads and fast dance songs. However, as shown by the black lines in the musical note hierarchy (left) and the dotted vertical lines in the AM hierarchy (right), the adjacent tiers of the hierarchy (i.e., green & blue and blue & red AM pairs) are dependent on each other compared with non-adjacent hierarchical relations (i.e., green-red AM pairing) and thus the hierarchy itself will expand or contract to fit the tempo.”

Comment 3:

The figures and descriptions of the findings (also in comparison to previous work) reveals that it would be interesting to extend the present set of analyses also to adult-directed speech in addition. This would allow for directly integrating the present work in previous approaches and allow for additional perspectives for future research. Relatedly, the authors discuss potential differences between different types of music (e.g., from different cultures with different underlying metric structures) as well as different languages (e.g., page 36 and elsewhere). It would be great to see how the models react to musical pieces in even versus uneven meters as well as to speech excerpts of different languages.

Response: Thank you for the comment. As suggested, we added ADS figure and also bird song figure to figure 1. Further, as indicated in the paper, we already published analyses of adult-directed speech (see Leong et al., 2017; Araujo et al., 2018). In this previous work examining the AM hierarchy of ADS, we demonstrated that ADS has significantly weaker phase synchronization between the slower bands of AMs centred on ~2 Hz and ~5 Hz compared to IDS. We also demonstrated that literacy affects phase synchronization. These prior analyses are discussed in the Introduction (see pages 9-10) when we motivate why we predict greater similarity between music and IDS than between music and ADS. We agree that in future work it would be interesting to apply the modelling to even versus uneven meters and to more languages. However, that is beyond the scope of the current paper. 

 

Comment 4:

Figure 2 suggests two slightly shifted peaks for music vs speech (e.g., with maxima of ˜1.5 vs. ˜2Hz?). Please add the exact maxima in the text.

Response: I thank the reviewers for the pertinent comments. As indicated, we described mean peak of music and IDS in the manuscript as follows. Further, we have also showed mean FFT results in S4 Appendix in both S-AMPH and PAD.

“The modelling showed that the AM bands in music matched those previously found in IDS, but the AM bands in the nature sounds did not. In particular, in panel 2d strong peaks in the delta and theta bands are clearly visible for instrumental music (red line, mean peak: delta 1.1Hz and 2.2Hz, theta 4.7Hz) and IDS (black line, mean peak: delta 1.8Hz, theta 3.3Hz), but not for nature sounds (blue line). Although the delta and theta peaks occur at slightly different temporal points, they are within close range of each other. Further, there are two matching peaks at delta and theta rates between IDS (black line in Fig 2d) and child song (light green in Fig 2d), but not in adult song, birdsong, and nature sounds.”

Comment 5:

The introduction includes a presentation of the Temporal Sampling theory. This section could be extended and clarified, notably how it links to other research domains focusing on the potential role of oscillations in this processing (whether typical or atypical). The explanation should be extended from the amplitude rise time of the vowel to that of a consonant, which contains the onset part and should be particularly relevant for the extraction of timing information. Regarding potential musical interventions and the discussion of amplitude envelopes of music, the authors should clarify the potential mechanisms that are boosted by the interventions.

Response: I thank the reviewers for all of the pertinent comments. However, prior research on rhythmic timing and amplitude rise times has already demonstrated that the rise time of the vowel is the key to speech rhythm, the consonants do not play a core role. Although a consonant before the vowel can move the temporal position of the perceived beat (e.g. sonorous consonant onsets produce later vowel peaks), the P centres literature already showed that the consonant in a syllable is not key to rhythmic timing (e.g., Scott 1991 showed that if two syllables with differing consonant onsets, like STREET and EAT are spoken to a rhythm, the rise time of the vowel governs syllable production). We now briefly mention P centres in the ms, noting potential mechanisms, please see page 7. 

Comment 6:

Regarding the window lengths selected by the modeler for the Allan Factor approach, it would be interesting to specify the time windows used (time scales) and whether this approach would work also for longer utterances? (page 7)

Response: I thank the reviewers for the comments. Allan factor analysis quantifies the clustering of events in terms of their variances in timing at different timescales. The time windows of a given size are tiled across a time series of events, and events are counted within each window. For example, in their study (Kello et al., 2017), recordings were chosen to be at least 4 min long, and window sizes were varied from approximately “15 ms to 15 s”. Their preliminary results showed that there was no need for windows shorter than 15 ms because events stopped being clustered, and 15 s is the largest window possible given a 4 min long recording. We now note their windows in our text (page 8).

Comment 7:

The authors did a great job in explaining TS, AM and brain oscillations (Page 12), but the explanations could be clarified by adding a figure illustrating the different elements (e.g., page 11).

Response: Thank you for the pertinent comments. We have adapted Fig 5 to meet this point. The AM bands (Fig 5 right) equate temporally to neural oscillatory rhythms. Human auditory rhythm perception relies in part on neural tracking of the AM patterns at different timescales simultaneously (e.g., green, blue, and red line in Fig 5). These temporal modulation patterns are then bound into a single sound percept. This neural tracking relies on acoustic components of the sound signal such as the amplitude rise times of nested AM components phase-resetting oscillatory cortical activity. We added the explanation in the legend of Fig 5 as follows.

“In the framework of Temporal Sampling theory, the AM bands (right) equate temporally to neural oscillatory rhythms. Auditory rhythm perception relies in part on neural tracking of the AM patterns at different timescales simultaneously (e.g., neural tracking of the green, blue, and red AMs in Fig 5 by neurophysiological delta, theta and alpha bands). This neural tracking is triggered by acoustic components of the sound signal such as the amplitude rise times (musical attack times) of the nested AM components which phase-reset oscillatory cortical activity.”

Comment 8:

Tal et al. (2017) addressing the ‘missing pulse’ phenomenon might be an interesting reference in the present context too. How do the present modeling approaches handle situations of the missing beat? Or more generally of syncopation and groove? The work by E. W. Large (University of Connecticut) proposes some interesting modeling perspectives for rhythm, meter and temporal information (as well as tonality), including the implication of neural oscillations as well as entrainment. It would be relevant for the present approach and for the integration in the research domain to also address these research approaches.

Response: I thank for letting us know these interesting papers. We have cited them in the Discussion and related them to our modelling work (see page 49). In our study, S-AMPH and PAD could detect missing beats as a silent gap (if no tone in music, no voice in speech). However, our modelling approach is basically only relevant to the part of Ed Large’s resonance theory that is based on the physical characteristics of the stimulus, as we now state on page 49.

“Introduction section: For music, oscillatory rhythms may align with rhythmic features of the acoustic input such as crotchets or musical beats (Doelling & Poeppel, 2015; Large et al., 2015; Di Liberto et al., 2020; Baltzell et al., 2019; Fujioka et al., 2015). However, possible correspondences between different oscillators and different musical units like crotchets and quavers have yet to be investigated.”

“Introduction section: Note finally that our modelling approach is conceptually distinct from models that identify the tactus or beat markers in singing (Coath et al., 2010), models of pulse perception based on neural resonance (Large et al., 2019), oscillatory models of auditory attention based on dynamic attending (Large & Jones, 1999), and models of temporal hierarchical structure based on the Allan Factor approach (Falk & Kello, 2017; Kello et al., 2017). Conceptually, ours is the only modelling approach to analyze the modulation structure of the amplitude envelope, recognized as core to speech processing by speech engineers (Greenberg, 2006). Our modelling decomposes the amplitude envelope and then relates the resulting AM bands and their phase relationships to individual musical units. In principle, this approach provides a novel acoustic perspective on musical rhythm, motivated by our prior novel acoustic analyses of Babytalk.”

“Discussion section: The modelling presented here also converges with past studies designed to detect pulse based on neural resonance theory (Large et al., 2019). Pulse is the perceptual phenomenon in which an individual perceives a steady beat. Large et al. (2019) suggested that the perception of pulse emerges through nonlinear coupling between two oscillatory networks, one representing the physical properties of the stimulus and a second network that integrates inputs from the sensory system. The nonlinear interactions between the two give rise to oscillatory activity not only at the frequencies present in the physical stimulus, but also at more complex combinations, including the pulse frequency. Consistent with this view, Tal et al. (2017) reported phase locking for the adult brain at the times of a missing pulse, even though the pulse was absent from the physical stimulus. This suggests that neural activity at the pulse frequency is (for adults) internally generated rather than being purely stimulus-driven. From this perspective, our modelling (i.e., S-AMPH and PAD) is capturing the physical stimulus characteristics (the modulation structure of the amplitude envelope and its internal phase relations) rather than capturing internally-generated oscillatory activity. To our knowledge, missing pulse phenomena have not yet been studied in infants. It may be that early learning of hierarchical phase relations from the amplitude envelopes of musical inputs may be required for the internal generation of missing pulse phenomena. On the other hand, ERP studies show that even newborns can detect beat violations in oddball paradigms, where occasionally a deviant rhythm with a missing downbeat is heard in place of a standard metrical rhythm (Winkler et al., 2009). Further studies with infants may also be able to investigate the phase relationships between missing pulses or beats and higher hierarchical units such as musical phrasing or prosody.”

Comment 9:

Methods

Page 15:

- Which are the “a priori assumptions” here?

Response: Thank you for the useful comment. The wording actually leads to misleading to readers. Therefore, we revised it as follows.

“The PAD model infers the modulators and a carrier based on Bayesian inference. PAD is biologically neutral and can be run recursively using different demodulation parameters each time to identify potential “priors” in the input stimulus.”

Comment 10:

Did the S-AMPH model use the same parameters as in Leong et al?

Response: Yes, the methodologies were based on a previous study by Leong and Goswami (2015). To establish the patterns of spectral modulation, the raw acoustic signal was passed through a 28 log-spaced ERBN filterbank spanning 100–7250 Hz. Further, the Hilbert envelopes of each of the spectral bands were passed through a 24 log-spaced ERBN filterbank spanning 0.9–40 Hz. This was specified in the begging of the section 2.2.1 (Signal Processing: Spectral and Temporal Modulations) as follows.

“The methodologies were based on a previous study by Leong and Goswami (2015).”

Comment 11:

Page 19:

- Does Figure 1 present just one example for music, IDS and machine? Could the authors propose a summary across items and show frequency ranges (in an additional figure?), aiming to provide information about the generalization of the observed pattern.

Response: I thank the reviewers for the pertinent comments. The methods used to generate Figure 1 require a representative piece of acoustic stimulus, not a summary. A summary across items is shown subsequently in Fig 2c and Fig2d. Also, we showed the summary for both S-AMPH and PAD in Table d in S4 Appendix. We cannot do something like averaging to generate Fig 1 because even same categorical sounds have in principle different stimuli although the temporal hierarchy is similar. That is, if they are averaged, the inherent characteristics of temporal hierarchy cancel each other out.

Comment 12:

Figure 1 proposes a comparative presentation of the four categories only for PAD. It would be informative for the reader to see the same type of presentation for S-AMPH.

Response: Thank you for the important comment. In this figure, we are showing how each sound statistically or acoustically includes a temporal hierarchy without the sensory/neural perspective imposed by the human cochlear and represented by S-AMPH. Because the S-AMPH models the cochlear filterbank, the frequency component between boundaries of the adjacent filterbanks (e.g., <0.9Hz, 2.5Hz, 7Hz, 17Hz, and 30Hz) is partially disappeared. This leads to inappropriate scalograms. We now state this in the Figure Legend. On the other hand, we added both ADS and bird song figure to figure 1 so that readers can compare IDS vs. ADS, other sounds and bird song. 

 

Comment 13:

Page 27: “suggestive of shared physical stimulus characteristics to which the brain can entrain” – So this would concern only the evoked responses (entrainment based on the input), can the authors also address entrainment based on cognitive construct (such as metrical hierarchies) that are not necessarily implemented in the acoustic signal?

Response: I thank the reviewers for the pertinent comments. This issue is now addressed in the new section concerning neural resonance theory (page 50), please see response to Comment 8. 

Comment 14:

Figure 3: How many items were used for each category here?

Response: Thanks for the comments. The sample size and amount of items in each category was described in the S1 Appendix in details (Corpora of Music, Speech, and Nature sound). We also stated it in the Materials and Methods section as follows.

“The sample size and number of items in each category is provided in S1 Appendix.”

Comment 15:

Discussion

Page 46: Please clarify the extension to music intervention. How could the findings be linked? Which type of music or characteristics should be used to train speech (across development as well as across different languages)?

Response. Our study may suggest that interventions utilising “Western” music that has temporal hierarchy known to correspond to English IDS (particularly delta rhythm and synchronization between delta and theta rhythms) may be beneficial for children with disorders of English language learning. As indicated, we revised the sentence as follows.

“The modelling presented here is also relevant to the remediation of childhood language disorders. The possible utility of musical interventions for children with disorders of language learning such as developmental language disorder (DLD) and developmental dyslexia has long been recognized (Ladányi et al., 2020; Cumming et al., 2015; Kodály, 1974; Jacques-Dalcroze, 1980). Such interventions are likely to be most beneficial when the temporal hierarchy of the music corresponds to the temporal hierarchy underpinning speech rhythm (Goswami, 2019a; Goswami, 2019b). Careful consideration of the statistical rhythm structures characterizing speech in different languages may thus lead to better remedial outcomes. For example, our findings suggest that for children with disorders of English language learning, interventions using Western music should be beneficial via the shared temporal hierarchy with English IDS and CDS. Further, it is possible that such interventions could be beneficial for second language learners.”

Comment 16:

Abstract:

Please clarify the wording, notably to which part of the sentence “which matched IDS” is referring to.

Response: Thank for the useful comment. As suggested, we clarified it in the Abstract section as follows.

“Both models revealed an hierarchically-nested AM modulation structure for music and song, but not nature sounds. This AM modulation structure for music and song matched IDS.”

Comment 17:

Introduction:

Page 4: this cognitive capacity has been shown to go beyond verbal material, that is, it is not restricted to language, but extends to non-linguistic materials, such as tones (e.g., Saffran et al., 1999; Tillman & Poulin-Charronnat, 2010), timbres (e.g., Loui et al., 2022; Tillman & McAdams, 2003; Tillman & Hoch, 2010) as well as rhythm and timing (e.g., Prince et al., 2018; Brandon et al., 2012).

Response: I thank the reviewers for the pertinent comments. We have now cited these important studies in the Introduction section as follows.

“Although language acquisition by human infants was once thought to require specialized neural architecture, studies of infant statistical learning have revealed that basic acoustic processing mechanisms are sufficient for infants to learn phonology (speech sound structure at different linguistic levels such as words, syllables, rhymes and phonemes; e.g. Saffran, 2001). Further, the cognitive capacity of statistical learning is not restricted to verbal language, but extends to non-linguistic sounds such as tones (e.g., Saffran et al., 1999; Francois et al., 2011), timbres (e.g., Loui et al., 2022; Tsogli et al., 2019) as well as rhythm and timing (e.g., Prince et al., 2018; Brandon et al., 2012; Daikoku et al., 2020).”

Comment 18:

Page 5: Does this depend on the language? Here the authors refer to their own work in English (Leong et al). It would be interesting to comment on extensions to other languages (e.g., German, French, Spanish, Italian) and whether it would be affected by differences between American English, British English, Australian ..?

Response: As suggested, we now include some discussion about other languages, other species, and music relationships in the Introduction section as follows.

“These phase relations between peaks and troughs in AM bands centred on ~2 Hz and ~5 Hz have also been revealed by statistical modelling of other languages like Portuguese and Spanish (Araujo et al., 2018; Pérez-Navarro et al., 2022). For example, Pérez-Navarro et al. (2022) reported that CDS in Spanish was characterized by higher temporal regularity of the placement of stressed syllables (phase synchronization of ~2 Hz and ~5 Hz AM bands) compared to ADS in Spanish. Further, phase relations are statistical characteristics that describe music as well as language, and phase relations appear relatively uniform regarding music from different cultures (Mehr et al., 2020; McPherson et al., 2020), as well as songs of different species (Roeske et al., 2020). Even prior to the acquisition of culture-specific biases of musical rhythm, infants are affected by ratio complexity (Hannon et al., 2011). Thus, phase hierarchies may be a universal aspect across music and language.”

Comment 19:

Page 13. A cortical tracking approach has been applied to music (even though not specifically to rhythm) by Pelofi, Shamma, and collaborators (see https://clame.nyu.edu/scientists/claire-pelofi), and which might be of interest for the authors here too.

Response: I thank the reviewers for letting me know the important researches. We have described and cited their papers in the manuscript as follows.

“For music, oscillatory rhythms may align with rhythmic features of the acoustic input such as crotchets or musical beats (Doelling & Poeppel, 2015; Large et al., 2015; Di Liberto et al., 2020; Baltzell et al., 2019; Fujioka et al., 2015). However, possible correspondences between different oscillators and different musical units like crotchets and quavers have yet to be investigated.”

“Further, Di Liberto and colleagues revealed that musical expertise increases the accuracy of cortical tracking (Di Liberto, Pelofi, Shamma, and de Cheveigné, 2020).”

Comment 20:

Page 30 “the nightingale song” – this suggests as if only 1 item was used here. Please clarify (and if yes, justify) or, preferably, extend to more items per category.

Response: I thank the reviewers for the pertinent comments. We actually analyzed 47 items for bird songs. However, as indicated, the wording sounds as if only 1 item was used here. We refined the wording as follows.

“at least for the corpus of the 47 nightingale songs (see, S1 Appendix) analyzed here.”

Comment 21:

Page 32: “When MI analyses were applied …..” Does this refer to the music material? Please clarify.

Response: Thanks for the helpful comment. We clarified it as follows.

“When MI analyses were applied for PAD in music, four peak frequencies were detected at ~2.4 Hz, ~4.8 Hz, ~9 Hz and 16 Hz.”

Comment 22:

Page 40 “similar to a harsh rhythmic whisper” For the interested reader, it would be great to add a sound file to the supplementary material section.

Response: I thank the reviewers for the comment. As suggested, we added it in the following link: https://osf.io/6s8kp/, and stated it in the manuscript as follows.

“The resulting percept was similar to a harsh rhythmic whisper. The sound is available from here https://osf.io/6s8kp/.”

Comment 23:

Page 42: The text refers here to “across Western musical genres and instruments” while most figures do not separate across genres. It would be helpful to see how findings potentially change as well as regarding whether the musical excerpts include voice or not.

Response: I thank the reviewers for the helpful comment. We showed each genres and instruments in S6 Appendix. But to make it simple and clear to the reader, we showed the figures of summary. As stated, we modified the sentence as follows.

“Indeed, the multi-timescale synchronization found here was systematic across Western musical genres and instruments (see S6 Appendix)”

Comment 24:

Appendix 1 lists the used materials - would the sound files be available upon (reasonable) request?

Response: All original infant-directed speech files are publicly available from the Figshare database: http://figshare.com/articles/SAMPH_CDS/1318572 DOI: 10.6084/m9.figshare.1318572. Please see the original article that used the speech data for more detailed information (Leong et al., 2015). All original bird song, nature sound files are available from https://www.xeno-canto.org/, https://mixkit.co/free-sound-effects/nature/, and https://www.zapsplat.com. Music and human song data has copyright, but described the detailed information in S1 Appendix. They are available by purchasing. We described about them in the section of “Data Availability Statement”.

---

## [Decision Letter · Decision Letter 1]

13 Jul 2022

PONE-D-21-38111R1Hierarchical Amplitude Modulation Structures and Rhythm Patterns: Comparing Western Musical Genres, Song, and Nature Sounds to BabytalkPLOS ONE

Dear Dr. Daikoku,

Thank you for submitting your manuscript to PLOS ONE. After careful consideration, we feel that it has merit but does not fully meet PLOS ONE’s publication criteria as it currently stands. Therefore, we invite you to submit a revised version of the manuscript that addresses the points raised during the review process.

The manuscript has been evaluated by one reviewer, and his comments are available below.

The reviewer has raised a number of concerns. He requests improvements to the reporting of methodological aspects of the study, for example, regarding the average duration across the different musical pieces used.  The reviewer also requests revision to the introduction and discussion.

Could you please carefully revise the manuscript to address all comments raised?

We look forward to receiving your revised manuscript.

Kind regards,

Lorena Verduci

Staff Editor

PLOS ONE

Journal Requirements:

Reviewers' comments:

Reviewer's Responses to Questions

**Comments to the Author**

1. If the authors have adequately addressed your comments raised in a previous round of review and you feel that this manuscript is now acceptable for publication, you may indicate that here to bypass the “Comments to the Author” section, enter your conflict of interest statement in the “Confidential to Editor” section, and submit your "Accept" recommendation.

Reviewer #2: (No Response)

2. Is the manuscript technically sound, and do the data support the conclusions?

Reviewer #2: Yes

3. Has the statistical analysis been performed appropriately and rigorously? 

Reviewer #2: Yes

4. Have the authors made all data underlying the findings in their manuscript fully available?

Reviewer #2: Yes

5. Is the manuscript presented in an intelligible fashion and written in standard English?

Reviewer #2: Yes

6. Review Comments to the Author

Reviewer #2: I thank the authors for their revision, which considerably improved the manuscript. I also welcome their “open-science” attitude. I just have the following clarification questions:

- In the introduction, the authors indicate that their modeling approach is “conceptually distinct from … models of pulse perception based on neural resonance (Large et al., 2019), oscillatory models of auditory attention based on dynamic attending”. In the discussion section, however, the authors discuss the convergence of their modelling approach with neural resonance theory. Please clarify. Also it is not clear why it is presented as being distinct from dynamic attending models as these also include multiple oscillators that entrain to the stimulus and influence processing, and also temporal sampling framework has been presented in link with dynamic attending (e.g., Goswami, 2011). Please clarify.

- Page 23: “The methodologies were based on a previous study by Leong and Goswami (2015).” Please clarify whether these were adapted (and “based on”/inspired by?) or whether the same implementations (i.e., same parameters, steps, etc.) were used here as in Leong and Goswami (2015). Otherwise, please indicate what was changed (and why).

- Thanks also for adding the various appendices for further information. Regarding appendix S1, could you clarify also the average duration across the different musical pieces used? I guess “Duration (minutes)” currently just indicates the total duration of all pieces together? Or are all pieces listed underneath played in their entirety ? (no excerpts chosen). Thanks.

7. PLOS authors have the option to publish the peer review history of their article (what does this mean?). If published, this will include your full peer review and any attached files.

Reviewer #2: No

---

## [Author Response · Author response to Decision Letter 1]

3 Aug 2022

RESPONSE TO REVIEWERS 2: 

I am extremely grateful for the Reviewers’ insightful comments on our manuscript. We have studied the Reviewers’ comments carefully and have made the necessary corrections in our paper. We believe that the comments have helped us significantly improve and refine the manuscript. Our responses to the Reviewers’ comments are as follows.

Comment 1:

In the introduction, the authors indicate that their modeling approach is “conceptually distinct from … models of pulse perception based on neural resonance (Large et al., 2019), oscillatory models of auditory attention based on dynamic attending”. In the discussion section, however, the authors discuss the convergence of their modelling approach with neural resonance theory. Please clarify. Also it is not clear why it is presented as being distinct from dynamic attending models as these also include multiple oscillators that entrain to the stimulus and influence processing, and also temporal sampling framework has been presented in link with dynamic attending (e.g., Goswami, 2011). Please clarify.

Response: We thank the reviewers for the pertinent comments. This made us reflect that our modelling is theoretically rather than conceptually distinct from other theories, while sharing some conceptual similarities with other approaches. Hence we reworded some of the sentences in the Introduction section and the Discussion section. The difference from dynamic attending theory is that DAT hypothesised multiple oscillators based on behavioural findings with attention tasks, rather than identifying a hierarchy of oscillators related to physical variations in stimuli. Here we specify the key oscillators in musical rhythm and the expected AM hierarchy a priori, based on our prior computational modelling of acoustic rhythm in language.

Introduction (pp. 15-16)

Note finally that our modelling approach is theoretically distinct from models that seek to identify the tactus or beat markers in singing (Coath et al., 2010), models of pulse perception based on neural resonance (Large et al., 2019), oscillatory models of auditory attention based on dynamic attending (Large & Jones, 1999), and models of temporal hierarchical structure based on the Allan Factor approach (Falk & Kello, 2017; Kello et al., 2017). Ours is the only modelling approach to analyze the modulation structure of the amplitude envelope and further to make specific a priori predictions concerning expected key temporal AM rates and key hierarchical AM phase relations related to the perception of musical rhythm structure and the parsing of musical units. We predict that the phase dependency between bands of AMs centred on ~2 Hz and ~ 5 Hz will relate to musical rhythm across different genres, and that music will show similar hierarchical AM structures in predictable spectral bandings to IDS, structures that can provide a perceptual basis for perceiving musical notes and musical phrasing. The amplitude envelope is recognized as core to speech processing by speech engineers (Greenberg, 2006). Our modelling decomposes the amplitude envelope of music instead of speech and then relates the resulting AM bands and their phase relationships to individual musical units. In principle, this approach provides a novel acoustic perspective on musical rhythm, motivated by our prior novel acoustic analyses of Babytalk.

Discussion (p. 51)

“The modelling presented here also converges conceptually with past studies designed to detect pulse based on neural resonance theory (Large et al., 2019). 

Comment 2:

- Page 23: “The methodologies were based on a previous study by Leong and Goswami (2015).” Please clarify whether these were adapted (and “based on”/inspired by?) or whether the same implementations (i.e., same parameters, steps, etc.) were used here as in Leong and Goswami (2015). Otherwise, please indicate what was changed (and why).

Response: Thank you for the comment. This study used the same methodologies as a previous study by Leong and Goswami (2015). We now stated it in the Methods section as follows.

“This study used the same methodologies and parameters as a previous study based on CDS by Leong and Goswami (2015) (for wiki, please see https://www.cne.psychol.cam.ac.uk).” (p. 23).

Comment 3:

- Thanks also for adding the various appendices for further information. Regarding appendix S1, could you clarify also the average duration across the different musical pieces used? I guess “Duration (minutes)” currently just indicates the total duration of all pieces together? Or are all pieces listed underneath played in their entirety ? (no excerpts chosen). Thanks.

Response: I thank the reviewers for the pertinent comments. All pieces listed underneath are played in their entirety. As suggested, we described the average duration as well in the S1 Appendix, as follows.

1) Single instrument materials

Instrument Representative

Composers Duration (minutes) Average daution (munutes) # pieces

Piano Beethoven, Mozart 230 3.97 58

Cello Bach,

Ysaÿe 173.8 3.78 46

Bass Bach 60.9 3.38 18

Viola Bach, Hindemith 320.8 4.65 69

Violin Bach,

Ysaÿe 206.5 4.39 47

Guitar Bach,

Sanz 218.4 3.36 65

2) Ensemble recordings

Genre Composers or

performers Duration

(minutes) Average daution (munutes) # pieces

Symphony Bach, Mozart, Beethoven 546.3 7.19 76

Jazz Miles Davis, Dave Brubeck 212.2 5.89 36

Rock The Beatles, U2 226.6 3.28 69

Children’s song (English) various 104.0 2.67 39

---

## [Decision Letter · Decision Letter 2]

22 Sep 2022

Hierarchical Amplitude Modulation Structures and Rhythm Patterns: Comparing Western Musical Genres, Song, and Nature Sounds to Babytalk

PONE-D-21-38111R2

Dear Dr. Daikoku,

We’re pleased to inform you that your manuscript has been judged scientifically suitable for publication and will be formally accepted for publication once it meets all outstanding technical requirements.

Kind regards,

Yann Benetreau, PhD

Division Editor

PLOS ONE

Additional Editor Comments (optional):

Reviewers' comments:

Reviewer's Responses to Questions

**Comments to the Author**

1. If the authors have adequately addressed your comments raised in a previous round of review and you feel that this manuscript is now acceptable for publication, you may indicate that here to bypass the “Comments to the Author” section, enter your conflict of interest statement in the “Confidential to Editor” section, and submit your "Accept" recommendation.

Reviewer #2: All comments have been addressed

2. Is the manuscript technically sound, and do the data support the conclusions?

Reviewer #2: Yes

3. Has the statistical analysis been performed appropriately and rigorously? 

Reviewer #2: Yes

4. Have the authors made all data underlying the findings in their manuscript fully available?

Reviewer #2: Yes

5. Is the manuscript presented in an intelligible fashion and written in standard English?

Reviewer #2: Yes

6. Review Comments to the Author

Reviewer #2: I thank the authors for the changes and latest additions. The manuscript has been further improved and the changes will clarify the work for the readers.

7. PLOS authors have the option to publish the peer review history of their article (what does this mean?). If published, this will include your full peer review and any attached files.

Reviewer #2: No

---

## [Editor Report · Acceptance letter]

26 Sep 2022

PONE-D-21-38111R2 

Hierarchical Amplitude Modulation Structures and Rhythm Patterns:
Comparing Western Musical Genres, Song, and Nature Sounds to Babytalk 

Dear Dr. Daikoku:

I'm pleased to inform you that your manuscript has been deemed suitable for publication in PLOS ONE. Congratulations! Your manuscript is now with our production department. 

Kind regards, 

on behalf of

Dr. Yann Benetreau 

Staff Editor

PLOS ONE